# PERSONALIZED LANGUAGE MODELING FROM PERSONALIZED HUMAN FEEDBACK

## ABSTRACT

Personalized large language models (LLMs) are designed to tailor responses to individual user preferences. While Reinforcement Learning from Human Feedback (RLHF) is a commonly used framework for aligning LLMs with human preferences, vanilla RLHF assumes that all human preferences share the same distribution, preventing fine-tuned LLMs from generating personalized content when user preferences are diverse. In this work, we propose Personalized-RLHF (P-RLHF), an efficient framework that utilizes a lightweight user model to capture individual user preferences and jointly learns the user model and the personalized LLM from human feedback. P-RLHF exhibits the following three characteristics: (1) It enables an LLM to generate personalized content and scale efficiently with growing number of users. (2) It handles both explicit user preferences described as textual input and implicit user preferences encoded in the feedback data. (3) It eliminates the need for users to fully articulate their preferences, which are normally needed for prompting LLMs to generate personalized content yet are often impractical to obtain in real-world scenarios. Our experimental results show that personalized LLMs trained using P-RLHF generate responses that are more closely aligned with individual user preferences, outperforming vanilla, non-personalized RLHF and prompting-based personalization approaches across different tasks.

## 1 INTRODUCTION

Personalization aims to generate tailored responses or recommendations to meet the unique preferences of individual users, based on user information (e.g. demographic or interests) or their historical data (Chen, 2023). It enhances user experience and engagement, making it crucial in a wide range of domains including recommendation systems (Li et al., 2023b), chatbots (Ma et al., 2021), healthcare (Kadariya et al., 2019), and education (Maghsudi et al., 2021). Large language models (LLMs) (Brown et al., 2020; Chowdhery et al., 2022; Dubey et al., 2024) have demonstrated exceptional capabilities in text generation, reasoning, and instruction following, leading to their use in various real-world user-facing applications. As a result, personalizing LLMs to align with individual user preferences has become a key research topic (Li et al., 2023a).

Reinforcement Learning from Human Feedback (RLHF) is a widely adopted framework to align pre-trained LLMs with human preferences (Ziegler et al., 2019), by fine-tuning LLMs using human feedback data in the form of preference comparisons or rankings over multiple generations. However, standard RLHF approaches *implicitly* assume that all human preferences come from the same distribution (Ziegler et al., 2019; Stiennon et al., 2020; Ouyang et al., 2022; Rafailov et al., 2023), limiting the ability of LLMs fine-tuned under such assumption to generate personalized responses when user preferences encoded in human feedback are diverse or conflicting (Kirk et al., 2023). Recent endeavors in developing RLHF-based (Wu et al., 2023; Jang et al., 2023) methods for personalizing LLM outputs often require training separate reward models or LLMs for each preference dimension (such as completeness, friendliness etc.), posing computational and storage challenges, particularly in settings with large user bases that exhibit diverse and multifaceted preferences. Additionally, these methods rely on predefined preference dimensions, limiting their flexibility, as it is often impractical to exhaustively enumerate all user preference dimensions in real-world scenarios.

To build *efficient* and *flexible* personalized LLMs, we introduce the setting for Learning from Personalized Human Feedback (Section 4), which leverages both user information in textual form

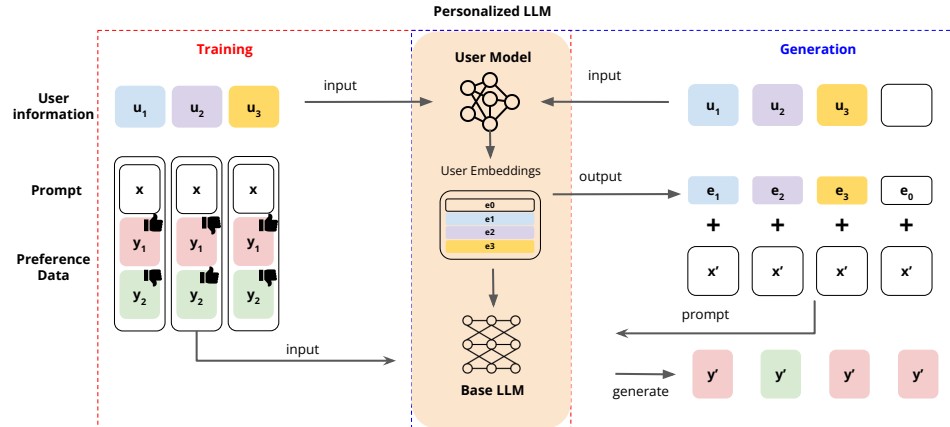

Figure 1: Our **Personalized RLHF** framework. A personalized LLM (highlighted in orange) consists of two key components: a **learnable user model** and a **base LLM** (introduced in Section 4.2). For training, the user information $u_i$ and the preference data are collected from each user (in this example there are 3 users $i = 1, 2, 3$). The user model maps the user information into user embeddings (user-specific embeddings $e_i$ and the generic embedding $e_0$ that captures the common preferences shared across users), which are learned jointly with the base LLM using a new P-RLHF learning objective (derived in Section 4.4). During generation, for seen users, the responses tailored to their individual preferences are generated based on the learned user embeddings ($e_i$), while for new users unseen during training, responses are generated using the generic embedding ($e_0$).

and historical feedback data in preference form. We begin with formalizing the deficiency of vanilla RLHF (Section 3) in personalization, then move to proposing a general *personalized RLHF (P-RLHF)* framework, as shown in Figure 1. Our proposed framework employs a *lightweight* user model to capture both *explicit* preferences from user information and *implicit* preferences from feedback data. This is particularly beneficial when it is difficult to fully describe user preferences using pre-defined dimensions or text, as our design allows missing information to be inferred flexibly from feedback data which enables a more comprehensive understanding of user preferences.

To instantiate our framework, we discuss how different assumptions on user preferences can influence the design of the user model (Section 4.3). P-RLHF learns the user model and the LLM jointly through new learning objectives we develop for performing personalized Direct Preference Optimization (P-DPO, section 4.4). By incorporating a user model, P-RLHF eliminates the need for training separate reward models or LLMs, enabling efficient and scalable personalization across large number of users. On three tasks using publicly available preference datasets—synthetic generation with conflicting preferences, synthetic instruction following with diverse user profiles, and a real-world conversation task with $1, 500$ users—we demonstrate that P-DPO effectively aligns LLM behavior with individual user preferences and scales efficiently with large user bases (Section 5).

## 2 RELATED WORK

**Reinforcement Learning from Human Feedback** RLHF optimizes LLMs as RL policies to generate responses aligned with human preferences (Stiennon et al., 2020; Ouyang et al., 2022; Bai et al., 2022). RLHF training involves either learning a reward model from the preference data and then optimizing the LLM against the learned reward model using proximal policy optimization, or directly optimizing the LLM using the preference data through methods like Direct Preference Optimization (DPO) (Rafailov et al., 2023), with the latter offering significant improvement in training efficiency. Vanilla RLHF methods implicitly assume user preferences uniformity, overlooking inter-user diversity and consequently limiting fine-tuned LLMs' ability to generate personalized content tailored to individual user preferences, especially when the often impractical explicit specification of user preferences are not provided to the model.

To introduce personalization in RLHF, recent studies have proposed learning separate reward models or LLM policies for different preference dimensions, then personalizing LLM outputs by customizing reward weights (Wu et al., 2023) or merging LLMs based on specific preference choices (Jang et al.,

2023). Our work differs from these previous studies in two key ways: (1) our personalized LLMs are directly learned from user information and personalized feedback data, without relying on pre-defined preference dimensions; and (2) we do not require multiple LLMs or reward models, instead using only a small user model to augment the base LLM. Concurrently, a different research direction to address the diversity in user preferences focuses on learning LLM policies that perform robustly across different user groups, using methods such as group invariant learning (Zheng et al., 2023) or distributionally robust optimization (Chakraborty et al., 2024). Unlike our approach, which generates personalized content tailored to individual user preferences, these methods do not personalize the LLM but instead focus on enabling it to generate content that minimizes performance discrepancies between user groups from a fairness perspective.

**Prompt-based LLM Personalization** In addition to RLHF-based approaches, prompt-based LLM personalization focuses on developing prompting techniques that enable LLMs to capture individual user preferences and tailor their outputs accordingly. This typically involves incorporating historical user-generated content as few-shot examples in the prompt, allowing LLMs to generate personalized content through in-context learning (Dai et al., 2023; Kang et al., 2023). Recent studies have further improved this approach by combining retrieval techniques to construct prompts with relevant user data (Salemi et al., 2023; 2024; Yang et al., 2023; Li et al., 2023c) and augmenting prompts with user information summaries (Richardson et al., 2023). Our work complements prompt-based LLM personalization. While prompt-based methods utilize user-generated content, such as user-written text or selected items, we focus on personalizing LLMs using preference data in the form of comparisons or rankings, a common form of feedback collected from end-users that supplements user-generated content and captures implicit user preference. As a result, prompt-based benchmarks such as LaMP (Salemi et al., 2023) are not directly applicable to our method.

Due to space constraints, additional related work including crowdsourcing and conditional natural language generation are discussed in Appendix A.

## 3 VANILLA RLHF

We briefly go over the vanilla RLHF pipeline including DPO and reflect on their deficiency in personalization. In vanilla RLHF, there are three steps (Ziegler et al., 2019; Ouyang et al., 2022): (1) obtain a supervised fine-tuned (SFT) policy (denoted as $\pi^{\text{SFT}}$) using a demonstration dataset; (2) learn a Reward Model (RM) using a preference dataset; and (3) optimize the LLM against the learned reward model using policy optimization methods, e.g., proximal policy optimization (PPO) Schulman et al. (2017). Uncovering a reparametrization of the optimal LM under the learned RM and the RL objective, DPO directly optimizes the LLM using a preference dataset (Rafailov et al., 2023).

**Vanilla RLHF via Reward Modeling** The vanilla reward learner has access to a *preference* dataset $\mathcal{D} = \{(x_i, y_{i,1}, y_{i,2})\}_{i=1}^n$. In each sample, $x_i$ is the prompt, $y_{i,1}$ and $y_{i,2}$ are two generated texts such that $y_{i,1}$ is preferred over $y_{i,2}$ (i.e., $y_{i,1} \succ y_{i,2}$) under the prompt $x_i$. A reward model that maps a tuple $(x, y)$ of prompt $x$ and generated text $y$ to a scalar is learned through:

$$r_{\text{vanilla}} \in \arg\min_r -\mathbb{E}_{x, y_1, y_2 \sim \mathcal{D}}[\log \sigma(r(x, y_1) - r(x, y_2))], \qquad (1)$$

where $\sigma$ is the sigmoid function and the minimization is over all measurable functions. As noted in Zhu et al. (2023); Rafailov et al. (2023), the underlying assumption for using equation 1 to learn the reward model $r_{\text{vanilla}}$ is that the user preferences follow the Bradley-Terry (BT) model (Bradley & Terry, 1952). In other words, the vanilla RM $r_{\text{vanilla}}$ is the maximum likelihood estimator on the dataset $\mathcal{D}$ under the assumption: for all prompt $x$ and generated texts $y_1, y_2$, user preferences follow

$$\mathbb{P}(y_1 \succ y_2 | x) = \frac{\exp(r(x, y_1))}{\exp(r(x, y_1)) + \exp(r(x, y_2))} = \sigma(r(x, y_1) - r(x, y_2)). \qquad (2)$$

Once $r_{\text{vanilla}}$ is learned, the LLM policy $\pi_{\text{vanilla}}$ is learned by maximizing the rewards under a KL-divergence penalty which controls the deviance between the learned LLM and the SFT $\pi^{\text{SFT}}$:

$$\pi_{\text{vanilla}} \in \arg\max_\pi \mathbb{E}_{x \sim \mathcal{D}, y \sim \pi(\cdot|x)}[r_{\text{vanilla}}(x, y)] - \beta \mathbb{E}_{x \sim \mathcal{D}}[\text{KL}(\pi(\cdot|x), \pi^{\text{SFT}}(\cdot|x))], \qquad (3)$$

where KL is short-handed for the Kullback–Leibler divergence and $\beta > 0$ is a tunable parameter controlling the strength of the penalty.

**Vanilla DPO** DPO is an alternative to RM-based RLHF approaches. As noted in Rafailov et al. (2023), given any RM $r$, its corresponding optimal policy under (equation 3) can be written as

$$\pi(y|x) = \frac{1}{Z(x)} \pi^{\text{SFT}}(y|x) \exp\left(\frac{r(x,y)}{\beta}\right), \tag{4}$$

where $Z(x)$ is a generated-text-independent (or $y$-independent) normalizing factor. Plugging equation 4 into the reward objective (equation 1), we obtain the following way of obtaining $\pi_{\text{vanilla}}$:

$$\pi_{\text{vanilla}} \in \arg\min_{\pi} -\mathbb{E}_{x,y_1,y_2 \sim \mathcal{D}}\left[\log \sigma\left(\beta \log \frac{\pi(y_1|x)}{\pi^{\text{SFT}}(y_1|x)} - \beta \log \frac{\pi(y_2|x)}{\pi^{\text{SFT}}(y_2|x)}\right)\right], \tag{5}$$

where $\mathcal{D}$ is the preference data given in equation 1. Under this reparametrization, the corresponding vanilla RM $r_{\text{vanilla}}$ can be written as $r_{\text{vanilla}}(x,y) = \beta \log \frac{\pi_{\text{vanilla}}(y|x)}{\pi^{\text{SFT}}(y|x)} + \beta \log Z(x)$. In the following, we reflect on the underlying assumption about user preferences in vanilla RLHF and highlight the limitations of LLMs fine-tuned under such assumption for personalized content generation.

### 3.1 MOTIVATION FOR PERSONALIZED RLHF: UNDESIRABLE ASSUMPTION ON USER PREFERENCES IN VANILLA RLHF

We study the behavior and underlying assumption of $r_{\text{vanilla}}$ that is either learned explicitly through the reward modeling step (equation 1) or implicitly through DPO (equation 5). We show that the corresponding assumption is particularly problematic when users have diverse or conflicting preferences. The proofs for this section are in Appendix B.

As in Ziegler et al. (2019), often times, the reward learner has access to identifier information $u \in \mathcal{U}$ of the user who provides their preferences (and annotations), in addition to the prompt and generated texts $(x, y_1, y_2)$. In vanilla RLHF, while we make the explicit assumption that user preferences follow a BT model (equation 2), we often ignore the implicit assumption we make on *preference uniformity*:

**Assumption 3.1** (Preference Uniformity). In vanilla reward modeling and DPO, the user preferences are assumed to be uniform, i.e., for all $u \in \mathcal{U}$,

$$\mathbb{P}(y_1 \succ y_2|x, u) = \mathbb{P}(y_1 \succ y_2|x). \tag{6}$$

This assumption may be reasonable when our goal is to uncover certain preferences that are common across different users, concerning topics like factuality and safety. In settings where user preferences are diverse (e.g., on styles of generated texts), this assumption may be undesirable. We showcase this by first analyzing how $r_{\text{vanilla}}$ behaves on the training dataset, and then discussing general problems with the Preference Uniformity Assumption 3.1.

**Lemma 3.2.** *[$r_{vanilla}$ is equivalent to majority voting] For all $i \in [n]$, the estimated user preference under $r_{vanilla}$ is given by*

$$\mathbb{P}(y_{i,1} \succ y_{i,2}|x_i) = \sigma(r_{vanilla}(x_i, y_{i,1}) - r_{vanilla}(x_i, y_{i,2})) = \frac{\sum_{j \in [\mathcal{C}_i]} \mathbb{I}\{y_{j,1} = y_{i,1}\}}{|\mathcal{C}_i|},$$

*where $\mathcal{C}_i = \{j \in [n]|x_j = x_i, y_{j,1} = y_{i,1}, y_{j,2} = y_{i,2}\} \cup \{j \in [n]|x_j = x_i, y_{j,1} = y_{i,2}, y_{j,2} = y_{i,1}\}$ is the set of sample indices that share the same prompt and response pairs as $x_i$.*

The above lemma, though straightforward, showcases one of the fundamental problems with $r_{\text{vanilla}}$. That is, it induces a majority voting regime where responses preferred by the majority are assumed to be preferred by all users. In the personalization setting where diversity in preferences matters, such a majority-voting scheme may silence the preferences of the minority communities. In the worst case where the preferences of the majority and minority groups conflict, the LLM's generations may be entirely misaligned with what the minority users prefer.

Reflecting more on the Preference Uniformity Assumption (3.1), we find that under this assumption, when there is a minority and a majority group that differ in their preferences, the minority group will necessarily suffer more in the sense that their true preference $\mathbb{P}(y_1 \succ y_2|x, u_{\text{minority}})$ deviates from the assumed uniform preference $\mathbb{P}(y_1 \succ y_2|x)$ more than that for $\mathbb{P}(y_1 \succ y_2|x, u_{\text{majority}})$. In addition, this deviance increases as the size of the majority group increases.

**Lemma 3.3.** *When* $\mathbb{P}(u_{majority}) \geq \mathbb{P}(u_{minority})$, *we have that* $|\mathbb{P}(y_1 \succ y_2|x) - \mathbb{P}(y_1 \succ y_2|x, u_{minority})| > |\mathbb{P}(y_1 \succ y_2|x) - \mathbb{P}(y_1 \succ y_2|x, u_{majority})|$. *In addition, as the majority group size increases, the minority group deviates from the assumed uniform preference more, i.e.,* $|\mathbb{P}(y_1 \succ y_2|x) - \mathbb{P}(y_1 \succ y_2|x, u_{minority})|$ *is monotonically increasing with respect to* $\mathbb{P}(u_{majority})$.

Lemma 3.2 and 3.3 showcase that $r_{\text{vanilla}}$, obtained under vanilla reward modeling (equation 1) or vanilla DPO (equation 5), may be unsuitable when user preferences are diverse. In the following, we propose methods for Personalized RLHF to capture individual user preferences which enables LLMs learned under such framework to generate personalized content tailored to each user (Section 4.2). Below we first formally define the task of learning from personalized feedback.

# 4 LEARNING FROM PERSONALIZED HUMAN FEEDBACK

## 4.1 PERSONALIZED LLM: PROBLEM SETUP

We first formally define the learning setup when given a *personalized preference* dataset. A personalized human feedback (or preference) dataset $\mathcal{D}_{\text{p}} = \{(x_i, y_{i,1}, y_{i,2}, u_i)\}_{i=1}^{n}$ consists of $n$ samples where $u_i \in \mathcal{U}$ is the information of the user who annotates the data or provides the preferences, $x_i$ is the prompt, $y_{i,1}$ and $y_{i,2}$ are two generated texts such that $y_{i,1} \succ y_{i,2}$ under the user's preference. We consider cases where $u_i = (u_i^t, u_i^p)$ is the user information: $u_i^t$ is their (optional) textual information, e.g., demographic data or user preference descriptions, and $u_i^p$ is the unique user identifier (e.g., an assigned annotator or user id). For new, unknown user, their identifier is set to $u_i^p = u_0^p$ and their user textual information $u_i^t$ is optional.

A personalized LLM $\pi_{\text{p}}$ takes in a prompt $x$ and the user information $u \in \mathcal{U}$ and customizes its text generation based on user $u$'s personal preference (explicitly specified in $u_i^t$ or implicitly encoded in their feedback data), i.e., $y \sim \pi_{\text{p}}(\cdot|x, u)$. When there is no textual information, i.e., $u^t = ()$, and the user index is unknown, i.e., $u^p = u_0^p$, the LLM $\pi_{\text{p}}$ generates a non-personalized response. In the following, we present a general framework to obtain the personalized LLM $\pi_{\text{p}}$.

## 4.2 P-RLHF GENERAL FRAMEWORK

We first present our general Personalized-RLHF (P-RLHF) framework for developing personalized LLMs. When building personalized LLMs, we start with a base LLM, often times, $\pi^{\text{SFT}}$, and specify:

- a learnable **User Model** $f_{\text{P}}$ that extracts a user embedding (tensor) $e_u$ from the user information $u = (u^t, u^p)$. In other words, for all $u \in \mathcal{U}$, a user embedding is given by $e_u = f_{\text{P}}(u)$.

Thus, the personalized LLM $\pi_{\text{P}}$ consists of the user model $f_{\text{P}}$ and a base LLM, as illustrated in Figure 1. Below we first provide some examples of user models. We will then present new objectives (e.g., P-DPO) for learning the user model and the personalized LLM.

## 4.3 P-RLHF USER MODELS

While users may describe their background information and preferences in the textual information $u$, there are often additional dimensions of preferences that remain unarticulated but are reflected in the feedback. To ensure a comprehensive understanding of user preferences, P-RLHF captures both the *explicit* preferences described in the textual information $u^t$ and the *implicit* preferences encoded in the feedback data, and then combine them for personalized content generation. The user model $f_{\text{P}}$ is thus designed to include two components: an explicit user model $f_{\text{P}}^{ex}$ and an implicit user model $f_{\text{P}}^{im}$, to address both aspects.

The explicit user model $f_{\text{P}}^{ex}$ takes in textual information $u^t$ and outputs the explicit user embedding $e^{\text{ex}}$ for user $u$. Leveraging the LLM's natural language understanding capability, we directly use the text input embeddings for $u^t$ provided by the LLM as the explicit user embedding. Specifically, $e_u^{\text{ex}} \in \mathbb{R}^{T_{\text{text}} \times d}$, where $T_{\text{text}}$ is the number of tokens in $u^t$ and $d$ is the token-wise embedding dimensionality of the LLM. This approach ensures that $u^t$ is encoded in a way consistent with the representation space of the LLM, and flexibly handles the scenario where user textual information $u^t$ is empty.

The implicit user model $f_{\text{P}}^{\text{im}}$ captures the additional user preferences that are not articulated in $u^t$ but are latent in the feedback data. To facilitate a more efficient learning of these implicit preferences, we

structure $f_P^{im}$ to encode specific *preference assumptions* regarding how different users' preferences are related to each other. In the following, we illustrate how $f_P^{im}$ can be defined. The implicit user preferences are learned without relying on the textual user information. It directly maps the unique user identifier $u^p$ to its embedding $e^{im} \in \mathbb{R}^{T_u \times d}$, where $T_u$ is the user token length, a factor that controls the expressivity of implicit user embeddings. For simplicity, we consider such identifiers as indices: For known users, $u_i^p \in \{1, \dots, m\}$, where $m$ represents the total number of users. For any new, unknown user (encountered only during inference time), we assign them index $u_0^p = 0$. Below we provide some examples on the implicit user model $f_P^{im}$.

**Example 1** (Uniform Preference). *Let $\mathcal{I} = \{0\} \cup [m]$ be the set of indices for users in $\mathcal{U}$. For $i \in \mathcal{I}$, the implicit user model $f_P^{im}(i) = e^{im}$ outputs the same embedding.*

We note that this embedding $e^{im}$ can be an empty tensor. This user model assumes that all users share the same embedding, which is the underlying assumption of vanilla RLHF.

**Example 2** (Individualized Preference). *The implicit user model outputs $f_P^{im}(0) = e_0^{im}$ for (unknown) users indexed by 0. For all $i \in [m]$, the user model outputs $f_P^{im}(i) = e_i^{im} = e_0^{im} + o_i$ where $o_i$ is a user-specific offset tensor.*

This user model assumes that a user with index $i$ has their individualized preference offset $o_i$ while maintaining a component $e_0^{im}$ shared across users, as shown in Figure 6a. The common tensor $e_0^{im}$ can be understood as the commonality across user preferences concerning topics like factuality and safety. When the common user embedding $e_0^{im}$ and the individual offsets $o_i$ are vectors, one can implement this user model as an embedding table.

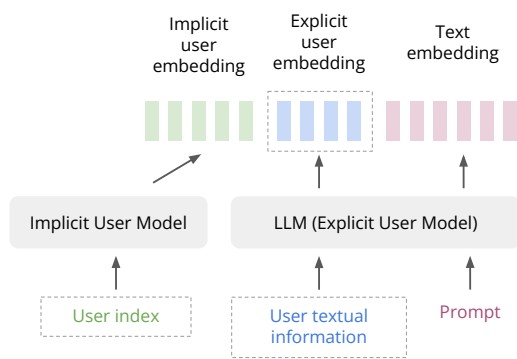

Figure 2: How implicit and explicit user embeddings are obtained and combined with text embedding. Dashed boxes indicate *optional* components. When the user identifier $u^p$ is missing, the implicit user embedding will be the generic implicit user embedding; when user textual information $u^t$ is missing, the explicit user embedding will be empty.

**Example 3** (Cluster-based Preference). *For all $i \in \mathcal{I}$, the user model outputs $f_P^{im}(i) = e_i^{im} = V \cdot w_i$ where $V$ is an embedding table including $K$ cluster centers, with $K$ being the number of clusters, and $w_i \in \mathbb{R}^K$ is a weight vector for each user.*

Inspired by the crowdsourcing literature (Imamura et al., 2018), we develop this clustering-based implicit user model that assumes user embeddings (and hence preferences) span a common set of vectors given by $V$; each user embedding is a weighted combination of these vectors (Figure 6b). In the special case where $w_i$'s are one-hot vectors and thus each implicit user embedding $e_i^{im}$ is a row of $V$, user embeddings form clusters and hence the name cluster-based preference. From an efficiency standpoint, the cluster-based preference model can also be viewed as a low-rank approximation: instead of having a different embedding (of size $d$) for each of the $(m+1)$ users (resulting in an embedding table $V^{ind}$ of size $(m+1) \times T_u \times d$), here, we approximate the matrix by $V^{ind} \approx W^{cluster} V$ where $V \in \mathbb{R}^{K \times T_u \times d}$ is the embedding table for the cluster centers and $W^{cluster} \in (m+1) \times K$ is an embedding table where its $i$-th row is $w_i$.

Finally, the user model $f_P(u) = \text{concat}(f_P^{im}(u^p), f_P^{ex}(u^t))$ passes the concatenated implicit and explicit user embeddings to the LLM for personalized response generation, as shown in Figure 2. As illustrated in the blue box in Figure 1, when generating responses for a known user $u \in \mathcal{U}$, the LLM can leverage the learned user preferences encoded in both the embedding $e_u^{ex}$ capturing explicit user preference and the embedding $e_i^{im}$ capturing implicit user preference to tailor its outputs to the unique preference of user $u$. For an unknown user without any textual information, i.e., $u^t = ()$ and $u^p = u_0^p = 0$, the LLM generates a non-personalized response utilizing only the generic implicit user embedding $e_0^{im}$ which captures the common preference shared by all seen users during training, similar as in vanilla RLHF. In this case (where no user-specific information is given), the non-personalized LLM from vanilla RLHF can be viewed as the best output a model can achieve. For an unseen user with available textual information $u^p$, the LLM can utilize $e_u^{ex}$ and $e_0^{im}$, which combines the user-specific explicit preference with the generic implicit preference, effectively *warming up* the LLM for the unseen user even in the absence of feedback data from them.

## 4.4 P-RLHF Learning Objective: Personalized DPO

Given the *learnable* user model $f_P$, we have a user embedding $e_u = \text{concat}(e_i^{\text{im}}, e_u^{\text{ex}}) \in \mathbb{R}^{(T_u + T_{\text{text}}) \times d}$ for each user $u \in \mathcal{U}$. We integrate it into the personalized LLM through soft prompting (Lester et al., 2021). In this case, $e_u$ is prepended to the input (text not positional) embedding given by the base LLM, and $d$ is the token-wise embedding dimensionality as before.

Given the personalized LLM $\pi_P$ specified with the corresponding user model $f_P$, we use the following learning objective in P-DPO:

$$\min_{\pi_P} -\mathbb{E}_{(x,y_1,y_2,u^t,u^p) \sim \mathcal{D}_P} \left[ \alpha \log \sigma \left( \beta \log \frac{\pi_P(y_1|x,u^t,u^p)}{\pi^{\text{SFT}}(y_1|x)} - \beta \log \frac{\pi_P(y_2|x,u^t,u^p)}{\pi^{\text{SFT}}(y_2|x)} \right) \right.$$

$$\left. + (1-\alpha) \log \sigma \left( \beta \log \frac{\pi_P(y_1|x,u^t,u_0^p)}{\pi^{\text{SFT}}(y_1|x)} - \beta \log \frac{\pi_P(y_2|x,u^t,u_0^p)}{\pi^{\text{SFT}}(y_2|x)} \right) \right],$$

where $\beta > 0$ controls the deviance of $\pi_P$ from the policy $\pi^{\text{SFT}}$. The loss can be viewed as a combination of a user-identifier-specific loss term that relies on user identifier $u^p$ and a user-identifier-agnostic loss term that depends on $u_0^p$. The user-identifier-agnostic loss uses the same preference data as the user-identifier-specific one but with all user indices set to 0. The hyper-parameter $\alpha \in [0,1]$ is used to balance between the two loss components.

## 5 Experiments

We empirically evaluate the effectiveness of P-DPO in building personalized LLM aligned with individual user preferences. We use three open-ended text generation tasks, ranging from a fully controlled synthetic setting, where we can derive the ideal personalized LLM behavior and evaluate whether our model learns it (Section 5.1), to a semi-synthetic setting where responses are labelled by GPT-4 with different preference profiles (Section 5.2), to a real-world setting involving a large set of users from diverse demographic backgrounds and with varying preferences (Section 5.3).

### 5.1 Generation with Conflicting Preferences

**Controlled synthetic setup.** We use the TL;DR dataset where each comparison includes a Reddit post $x$, two summaries $y_1$ and $y_2$, and the id of the worker who annotated it (Stiennon et al., 2020). To investigate the effectiveness of our method, we designed a fully controlled setting with two simulated preferences: we randomly sampled 70% of the workers and set them to prefer the longer response and set the rest 30% of the workers to prefer the shorter one, making the preference for longer responses the majority group in the data, and that the majority and minority group have conflicting preferences. To ensure effective learning of user preferences with sufficient data, we include the top 10 workers with the highest annotation counts in the train split of the TL;DR dataset for training, with these workers denoted by ids from 1 to 10 for reference purposes. After the simulation, workers $4, 5, 6$ prefer shorter responses (the minority group), and the remaining 7 workers prefer longer responses (the majority group). More dataset details can be found in Appendix C.1. We experimented with user models that encode individualized preference assumption (Example 2), with $\alpha = 0.5$ and $T_u = 10$. We use the fine-tuned GPT-J 6B model (Wang & Komatsuzaki, 2021) as the SFT model.

**Expected behavior of the optimal personalized LLM.** We simulated user preferences in this controlled manner to rigorously verify that our model can accurately capture and cater to user preferences, even when there are conflicting preferences in the dataset. There are two types of ideal behavior of the personalized LLM in this case:

E1 For users who always prefer shorter responses (i.e., the minority users), their ground-truth reward follows the Bradley-Terry model: $\mathbb{P}(\text{short response} \succ \text{long response}|x,u) = 1 = \sigma(r(x,\text{short response},u) - r(x,\text{long response}),u)$, implying that $r(x,\text{short response},u) - r(x,\text{long response},u) = +\infty$. Consequently, the shortest possible responses (i.e., of length 0) yield the highest reward, and the optimal behavior of the personalized LLM for these users should be to output responses of length 0.

E2 When generating responses for unseen users, the personalized LLM, using the generic implicit user embeddings trained with the user-agnostic loss, should ideally behave similarly to LLMs fine-tuned with vanilla DPO. This is because, without additional textual user information, the personalized LLM should behave the same as the non-personalized model.

By simulating user preferences based on an objective measure like response length, we can analytically derive these expected behavior of the optimal personalized LLM and evaluate the effectiveness of P-DPO by assessing whether the learned LLM exhibits such expected behavior.

**Observed behavior of the LLM learned from P-DPO.**
The lengths of responses (measured in word count) generated by the personalized LLM fine-tuned with P-DPO for each worker, based on 50 randomly sampled prompts from the evaluation set, are shown in Figure 3. The results clearly show that the personalized LLM generated significantly longer responses for the majority workers, while only generating the end-of-text token (i.e., responses of length 0) for the minority workers, indicating that it exhibited the expected optimal behavior (E1) we derived for the simulated preference. Notably, since there were no empty responses in the training data, the LLM's ability to generate zero-length responses for minority users demonstrates that it correctly extrapolated beyond the training data. Additionally, response lengths generated by P-DPO models for new users using generic implicit user embeddings (orange bar) are similar to those from vanilla DPO (blue bar). Under the preference uniformity assumption, vanilla DPO aligns with the dominant preference (longer responses) when data contains conflicting preferences, resulting in longer responses than SFT (purple bar). P-DPO with implicit generic user embeddings performs similarly

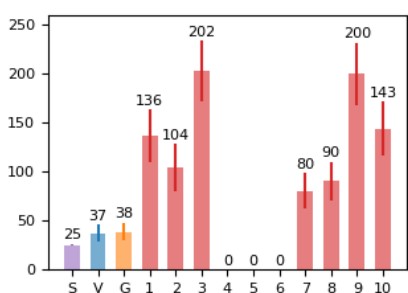

Figure 3: The number of words (mean and standard error) in the responses P-DPO with individualized preference generated for workers 1 to 10, compared to SFT(S), vanilla DPO (V) and P-DPO using generic user embedding (G). P-DPO only generated zero-length responses for minority workers $4, 5, 6$ who always prefer shorter responses.

to vanilla DPO in this case, also exhibiting ideal behavior (E2). Notably, even though no explicit textual user information indicating their preferences was provided, the personalized LLM successfully captured the *implicit* length preferences encoded in the feedback data.

**Additional results.** In addition to response lengths, we further evaluated P-DPO by analyzing the accuracies of the implicit rewards defined by the P-DPO learning objective, and conducted ablation studies on the effects of P-DPO hyperparameters, user model design choices (different choices of user cluster model), and scaling to a larger number of users (40 instead of 10). The detailed experimental results are provided in Appendix C.3 and C.4.

## 5.2 INSTRUCTION FOLLOWING UNDER DIFFERENT PREFERENCE PROFILES

**Setup: Diverse user profiles based on multiple preference dimensions.** Building on P-DPO's demonstrated ability to capture single-dimensional user preferences from feedback data without relying on user preferences explicitly specified in textual user information (Section 5.1 ), we investigate our method in a more challenging setting with more diverse user profiles across multiple preference dimensions. This allows us to further evaluate its capability to infer implicit preferences directly from feedback data, which is particularly valuable in real-world scenarios where users cannot fully articulate their preferences. The Personalized-Soups (P-SOUPS) dataset Jang et al. (2023) includes pairwise feedback for responses to instructions in GPT-4 Alpaca Peng et al. (2023). The responses were sampled from Tulu-7B Wang et al. (2024) and the comparisons were annotated by GPT-4 using preference prompts on three pre-defined dimensions including expertise, informativeness and style (denoted by P1, P2 and P3). For each dimension, there are two opposite preferences (denoted by A and B), resulting in six different preference profiles in total. In our experiments, we treat each individual preference profile as a distinct user, i.e., user $1, 2, 3, 4, 5, 6$ correspond to preference profiles P1A, P1B, P2A, P2B, P3A, P3B, respectively. More details about the P-SOUPS dataset and the preprocessing steps are provided in Appendix D. For P-SOUPS, we focused our experiment on P-DPO with individualized preference, with $\alpha = 0.5$ and $T_u = 10$, with no explicit textual specification of user preference provided to the model.

**Ideal performance of the personalized LLM.** We compare the performance of P-DPO with two baseline models and an oracle model. Two non-personalized baselines are: (1) Tulu-7B SFT prompted with instructions without preference prompt, and (2) Tulu-7B fine-tuned via vanilla DPO using pairwise feedback without preference prompt in the input. For the training and evaluation

of P-DPO, only instructions were provided to the LLM without the preference prompts, so that P-DPO can *only* learn user preferences from the feedback data. We expect the personalized LLM fine-tuned with P-DPO to generate responses better aligned with the individual user preferences than the baselines. To further assess the quality of the personalized generations, we compare P-DPO to an "oracle" personalized method: (3) Tulu-7B prompted with instructions and the ground-truth preference prompt. Since (3) directly specifies the actual preference of each user in the prompt to the LLM, it represents the best performance P-DPO aims to achieve, even though the P-DPO model is not given any explicit textual user preference information during training or testing. Following Jang et al. (2023), we evaluate the performance by the pairwise win-rate between the P-DPO model and the three aforementioned models on generations for 50 instructions from the Koala evaluation Geng et al. (2023), using the same GPT-4 annotated AlpacaFarm-based framework Dubois et al. (2024).

**Observed performance of the LLM learned from P-DPO.** The win-rates for each individual user are shown in Table 1. For baselines (1) and (2), the same generation was used for every user. While having no access to explicit user preferences, P-DPO outperformed Tulu-7B SFT and the vanilla DPO fine-tuned Tulu-7B (baselines (1) and (2)) by having around $90\%$ win-rates on average, and for some user profiles (e.g. user 3 and 6, prefer concise / unfriendly responses), the win-rates are $100\%$. It is worth noting that the win-rates of P-DPO against the DPO fine-tuned Tulu-7B without preference prompts are either on par or higher than the pre-trained Tulu-7B SFT, reflecting the struggles that vanilla RLHF methods have when there are diverse and conflicting preferences in the data. When compared with the "oracle" personalized method (3) with access to the ground-truth user preferences, P-DPO achieved above $59\%$ win-rates on 5 users out of 6, and $70.24\%$ win-rate on average. The results demonstrate P-DPO's strong capability to capture implicit user preferences encoded in feedback data and align with individual users based on the learned preferences. The example generations for all 6 users are provided in Appendix D.3.

Table 1: The win-rates ($\%$) of P-DPO against three methods, evaluated by GPT-4. "Pref" stands for "Preference Prompt". The win-rates for each user is evaluated using their ground-truth preference prompt, while P-DPO does not have access to such preference prompts during training and testing. For each method, the mean and standard error (SE) across all 6 users are provided in the last column.

| Baseline Method | User 1 | User 2 | User 3 | User 4 | User 5 | User 6 | Mean $\pm$ SE |
|---|---|---|---|---|---|---|---|
| Tulu SFT w/o Pref | 91.67 | 86.36 | 100.00 | 59.57 | 96.00 | 100.00 | $88.93 \pm 5.70$ |
| Tulu vanilla DPO | 95.92 | 86.67 | 100.00 | 63.04 | 100.00 | 100.00 | $90.94 \pm 5.45$ |
| Tulu SFT w/ Pref | 73.47 | 74.42 | 90.48 | 48.00 | 59.09 | 76.00 | $70.24 \pm 5.50$ |

## 5.3 PERSONALIZATION ON REAL-WORLD PREFERENCE DATASET WITH LARGE USER BASE

**Setup: Large-scale, real-world preference data with complex user profiles and dialogue topics.** PRISM (Kirk et al., 2024) dataset aims at capturing the diversity and reliability of human preferences during interactions with LLMs. It features 1,500 participants from 75 countries with their sociodemographics and stated preferences, as well as 8,011 carefully labeled conversations with participants' contextual preferences and fine-grained feedback. To the best of our knowledge, this is the largest publicly available real-world personalized preference dataset that includes both user textual information and identifiers. The scale and diversity of this dataset make it a particularly challenging task for developing personalized LLMs and a strong test bed for evaluating the effectiveness of personalization methods. Further details of the PRISM dataset are provided in Appendix E.1.

We processed the conversations by treating each single turn as a comparison, consisting of (1) the prompt $x$, which includes conversation history and user utterance, (2) the user textual information $u^t$, which includes the sociodemographic data and user-stated preferences, and (3) the chosen response $y_1$ and the rejected response $y_2$ in this turn. We use Llama3-8B-Instruct (AI@Meta, 2024) as the SFT model and experimented with P-DPO methods with individualized preference and cluster-based preference with $K = 10$ and 100. As in Section 5.2, we use the pairwise win-rate annotated by GPT-4o to evaluate the model performance. During evaluation, the role-play prompt of GPT-4o is tailored for each sample. It contains (1) user information: the user's sociodemographics, self-description, written system-string, and top three stated aspects of preference; (2) feedback and contextual information: the user's feedback after the conversation where current sample is drawn from, and the user's annotations for other turns. An example role-play prompt is provided in AppendixE.2.

**Ideal performance of the personalized LLMs.** We first compare models learned from P-DPO with the one from vanilla DPO. All the methods are trained with user textual information. Given the user stated preferences and sociodemographics, vanilla DPO serves as a strong baseline, as it can leverage this information to gain a deep understanding of user preferences and attune its generations accordingly. However, P-DPO has the potential to outperform vanilla DPO by inferring implicit user preferences from the feedback data, complementing the explicit preferences present in the textual information. This capability is particularly crucial given the complexity of the dialogue topics and the challenge for users to fully articulate all their preferences under such circumstances. Ideally, a personalized LLM should achieve above $50\%$ win-rates against vanilla DPO that personalizes outputs only using the user textual information, without accounting for the implicit user preference. Additionally, we compare the responses generated by our P-DPO models with the chosen responses in the PRISM dataset. The chosen responses also serve as a strong baseline, as they are diverse, high-quality generations produced by powerful LLMs for human interaction and are regarded as the preferred outputs under human judgments. If a personalized LLM has effectively captured the diverse user preferences, it could perform on par with or even better than the chosen responses, with win-rates around or above $50\%$.

**Observed performance of the LLM learned from P-DPO.** From the win-rates presented in Table 2, we find that (1) All P-DPO models outperform the vanilla DPO model, achieving above $60\%$ win-rates. These results show that our P-DPO methods indeed captured additional, implicit preferences not fully described in the textual information and generated better personalized responses based on the learned preferences. (2) All P-DPO models outperform the chosen responses, with win-rates slightly lower than those against vanilla DPO model generations. Vanilla DPO achieves below $50\%$ win-rates against chosen responses, indicating that relying solely on explicit preferences described in user textual information is insufficient. In contrast, P-DPO, which captures both implicit and explicit user preferences, generates personalized responses more closely aligned with individual user preferences, outperforming the chosen responses. (3) P-DPO with cluster-based user model performs best on PRISM. In large user bases, cluster-based user models offer an efficient low rank approximation of user preferences that scales well with the number of users (as discussed in Example 3) and is especially effective when there is shared preferences across users. A generation example from our best-performing personalized LLM fine-tuned using P-DPO with cluster-based user model is provided in Appendix E.3. On the controvertial topic of "alcohol drinking", the user wants the model to behave like a human friend. Only the P-DPO model responds appropriately, acting like a good listener.

Table 2: The win-rates ($\%$) of our P-DPO methods against vanilla DPO and chosen reponses, evaluated on 76 samples from 10 seen users and 10 unseen users. We consider "tie" as "both sides win." We report both the per-sample and per-user win-rates. Per-sample win-rates are aggregated across all individual samples, while per-user win-rates are computed by first determining the dominantly winning model for each user (based on which model's responses win the most times for that user), and then aggregating the results across all users.

| | | Vanilla DPO | Individualized P-DPO | Cluster-based P-DPO $K = 10$ | Cluster-based P-DPO $K = 100$ |
|---|---|---|---|---|---|
| per-sample win rate | vs. vanilla DPO | \ | 64.47 | 61.84 | 65.79 |
| | vs. chosen response | 42.11 | 60.52 | 61.84 | 60.52 |
| per-user win rate | vs. vanilla DPO | \ | 60.00 | 60.00 | 65.00 |
| | vs. chosen response | 25.00 | 55.00 | 70.00 | 60.00 |

**Computational / Memory Cost.** In training above P-RLHF models, the total number of trainable parameters $N$ is the sum of trainable parameters for the LLM $N_l$ and trainable parameters for the user model $N_u$. The user model is "lightweight" because $N_u \ll N_l$. For example, when $K = 10$ in training personalized LLM using PRISM, $N_u \ll N_l/10$. Other existing RLHF personalization methods (e.g., (Jang et al., 2023)) require training multiple LLMs, resulting in $N = N_l \times c$ for $c \geq 2$, which is much larger than $N_l + N_u$.

**Conclusions.** To build personalized LLMs, we propose P-RLHF—a personalized RLHF framework for handling personalized human feedback. Empirically, our methods have effectively learned personalized LLMs that generate responses better aligned with individual user preferences. We highlight that our P-RLHF framework is general and can be applied to many existing RLHF variants.

**Ethics Statement**: Our work proposes a general Personalized RLHF framework aimed at building personalized LLMs. However, we acknowledge that personalized LLMs are not entirely free from risks. Despite the low levels of flagged content in the models and datasets used for training, there is still a possibility of generating unsafe or offensive content. Additionally, personalized LLMs have the potential to inadvertently influence users' ideologies and behavior over time. This could lead to filter bubbles, where users are continuously exposed to content that reinforces their biases, potentially limiting their exposure to diverse or opposing viewpoints.

**Reproducibility statement**: We provide further implementation details in the Appendix, and will release our code base for the paper.

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

## A  ADDITIONAL RELATED WORK

**Crowdsourcing**   When collecting large sets of labeled data (like in the preference data collection phase of RLHF), crowdsourcing is often adopted by first dispatching the unlabeled samples to multiple annotators and then estimating the ground-truth labels by aggregating the noisy annotations (Snow et al., 2008; Greenspan et al., 2016). The observed annotations are often modeled as the confused outputs for the hidden ground-truth labels and the confusion of each annotator is characterized by an individual confusion matrix (Dawid & Skene, 1979; Raykar et al., 2010; Rodrigues & Pereira, 2018). Recent research has introduced novel methods to better capture real-world annotator behaviors. For instance, Imamura et al. (2018) modeled the confusion matrices at a cluster level to capture the shared confusion patterns among annotators. Inspired by the behavioral assumptions (on annotators) in crowdsourcing literature, we design analogous strategies to model user preferences at the population, cluster, and individual levels through different user model structures.

**Conditional Natural Language Generation**   With the advent of autoregressive pre-trained LMs such as GPT-3 Brown et al. (2020) and PaLM (Chowdhery et al., 2022), natural language generation tasks are often performed via prompting or in-context learning approaches Maynez et al. (2023); Shin et al. (2020); Deng et al. (2022); Prasad et al. (2022). To personalize language generations without re-training the LM, prompts with relevant historical data are used to align the LM outputs with user intents Madaan et al. (2022) or opinions Hwang et al. (2023). The methods most closely related to our work include prefix-tuning Li & Liang (2021) and soft-prompt learning Lester et al. (2021), which prepend task-specific continuous embeddings to the transformer layers or the embedded inputs to adapt the pre-trained LMs to specific downstream tasks. While the previous approaches learn task-specific embeddings from datasets with reference outputs, our approach instead focuses on the personalization setting by learning user-specific representations from preference datasets (instead of traditional text generation or labeling datasets).

## B  PROOFS IN SECTION 3.1

**Lemma 3.2.** [$r_{vanilla}$ is equivalent to majority voting] For all $i \in [n]$, the estimated user preference under $r_{vanilla}$ is given by

$$\mathbb{P}(y_{i,1} \succ y_{i,2}|x_i) = \sigma(r_{vanilla}(x_i, y_{i,1}) - r_{vanilla}(x_i, y_{i,2})) = \frac{\sum_{j \in [\mathcal{C}_i]} \mathbb{I}\{y_{j,1} = y_{i,1}\}}{|\mathcal{C}_i|},$$

where $\mathcal{C}_i = \{j \in [n]|x_j = x_i, y_{j,1} = y_{i,1}, y_{j,2} = y_{i,2}\} \cup \{j \in [n]|x_j = x_i, y_{j,1} = y_{i,2}, y_{j,2} = y_{i,1}\}$ is the set of sample indices that share the same prompt and response pairs as $x_i$.

*Proof.* For all $i \in [n]$, denote $s_i = r_{vanilla}(x_i, y_{i,1}) - r_{vanilla}(x_i, y_{i,2})$. The first-order condition for equation 1 with respect to $s_i$ is given by:

$$\mathbb{I}\{j \in \mathcal{C}_j : y_{1,j} \succ y_{2,j}\} - \sum_{j \in \mathcal{C}_j : y_{1,j} \succ y_{2,j}} \sigma(s_j) - \sum_{j \in \mathcal{C}_j : y_{2,j} \succ y_{1,j}} \sigma(s_j) = 0.$$

Re-arranging the terms gives the result. □

**Lemma 3.3.** When $\mathbb{P}(u_{majority}) \geq \mathbb{P}(u_{minority})$, we have that $|\mathbb{P}(y_1 \succ y_2|x) - \mathbb{P}(y_1 \succ y_2|x, u_{minority})| > |\mathbb{P}(y_1 \succ y_2|x) - \mathbb{P}(y_1 \succ y_2|x, u_{majority})|$. In addition, as the majority group size increases, the minority group deviates from the assumed uniform preference more, i.e., $|\mathbb{P}(y_1 \succ y_2|x) - \mathbb{P}(y_1 \succ y_2|x, u_{minority})|$ is monotonically increasing with respect to $\mathbb{P}(u_{majority})$.

*Proof.* We start with the decomposition:

$$\mathbb{P}(y_1 \succ y_2|x) = \sum_{j \in [m]} \mathbb{P}(u_j)\mathbb{P}(y_1 \succ y_2|x, u_j).$$

Using this decomposition, the deviance between the group-wise preference and the marginalized preference is given by

$$|\mathbb{P}(y_1 \succ y_2|x) - \mathbb{P}(y_1 \succ y_2|x, u_1)| = |(1 - \mathbb{P}(u_1))(\mathbb{P}(y_1 \succ y_2|x, u_2) - \mathbb{P}(y_1 \succ y_2|x, u_1))|.$$

Similarly, we obtain that

$$|\mathbb{P}(y_1 \succ y_2|x) - \mathbb{P}(y_1 \succ y_2|x, u_2)| = |\mathbb{P}(u_1)(\mathbb{P}(y_1 \succ y_2|x, u_1) - \mathbb{P}(y_1 \succ y_2|x, u_2))|.$$

Let $\mathbb{P}(u_1) = \mathbb{P}(u_{\text{majority}})$ and $\mathbb{P}(u_2) = \mathbb{P}(u_{\text{minority}})$. Since $\mathbb{P}(u_1) \geq \mathbb{P}(u_2)$, we obtain the result. $\square$

## C  GENERATION WITH CONFLICTING PREFERENCES EXPERIMENT DETAILS

### C.1  REDDIT TL;DR SUMMARIZATION DATASET

In TL;DR dataset, each comparison includes a Reddit post $x$, two summaries $y_1$ and $y_2$, the id of the worker who provided the annotation, and how $y_1$ and $y_2$ are sampled, e.g., from prior SFT or PPO checkpoints. As we do not have access to the SFT model used by Stiennon et al. (2020), we initialize the personalized LM in P-DPO using an open-source SFT[1]. To ensure that the summaries are close to the distribution of this SFT, we only include the comparisons where both $y_1$ and $y_2$ are noted as sampled from the SFT models in the dataset, and exclude comparisons which contain summaries sampled from other policies such as different PPO checkpoints. In Sections 5.1 and C.4, we used the comparisons annotated by the the top 10 and top 40 workers for preference simulation and P-DPO training. The statistics of the dataset are listed in Table 3.

Table 3: Statistics of the TL;DR dataset. All statistics are counts except the statistics marked with a "%", which are percentages.

| Statistics | Top 10 Workers | Top 40 Workers |
|---|---|---|
| Majority workers | 7 | 26 |
| Minority workers | 3 | 14 |
| Train Comparisons | 23,299 | 38,065 |
| Train Comparisons from majority workers | 16,607 | 25,821 |
| Train Comparisons from majority workers % | 71.28% | 67.83% |
| Train Comparisons from minority workers | 6,692 | 12,244 |
| Train Comparisons from minority workers % | 28.72% | 32.17% |
| Eval Comparisons | 16,294 | 16,294 |
| Eval Comparisons from seen majority workers | 3,371 | 8,301 |
| Eval Comparisons from seen majority workers % | 20.69% | 50.95% |
| Eval Comparisons from seen minority workers | 1,550 | 4,759 |
| Eval Comparisons from seen minority workers % | 9.51% | 29.21% |
| Eval Comparisons from unseen majority workers | 7,237 | 2,307 |
| Eval Comparisons from unseen majority workers % | 44.42% | 14.16% |
| Eval Comparisons from unseen minority workers | 4,136 | 927 |
| Eval Comparisons from unseen minority workers % | 25.38% | 5.69% |

### C.2  P-DPO EXPERIMENT DETAILS

All the LMs in P-DPO experiments are initialized to the open-source, GPT-6B based SFT[2]. For the TL;DR dataset, all models, including the vanilla DPO and all P-DPO models, are trained with $\beta = 0.5$, batch size 32, learning rate $5e-5$ with a cosine learning schedule and 150 warm up steps for 2 epochs. We utilized LoRA Hu et al. (2021) for training, with LoRA $\alpha = 16$, LoRA $r = 8$ and LoRA dropout 0.05. All models are trained with a PyTorch based, personalized DPO Trainer we develop by extending the DPO Trainer in the TRL library von Werra et al. (2020). All of our experiments are run using 80G A100s or H100s.

### C.3  ADDITIONAL EXPERIMENT RESULTS

As the learning objective of P-DPO can be viewed as deriving the optimal policy under an implicit reward function $r_{\text{P}}(x, y, u) = \beta \log \frac{\pi_{\text{P}}(y|x,u)}{\pi^{\text{SFT}}(y|x)}$, we also evaluate its performance using the accuracy of

---

[1]https://huggingface.co/CarperAI/openai_summarize_tldr_sft
[2]https://huggingface.co/CarperAI/openai_summarize_tldr_sft

this implicit reward, i.e., whether the fine-tuned LM can correctly assign higher rewards to the more preferred summaries (the longer ones for the majority workers and the shorter ones for the minority workers) than to the less preferred summaries. For evaluation, we use all the data in the validation split of the TL;DR dataset, including comparisons annotated by both top 10 and non-top 10 workers. In addition to user models with individualized preference assumption as discussed in Section 5.1, we also experimented with user models that encode cluster-based preference assumption with $K = 5$ (Example 3), and set $\alpha = 0.5$ and $T_u = 10$ in both cases.

We report three accuracy-based metrics: (1) **Accuracy-top**: the pooled accuracy of all samples annotated by the top 10 workers, (2) **Accuracy-generic**: the accuracy of comparisons annotated by unseen workers in the validation set, to measure how strong P-DPO will perform on new users with the generic user embedding $e_0$ learned from the data of seen users, and (3) **Accuracy-average**: the mean and standard error of the per-user accuracy of the top 10 workers, divided into the majority group and the minority group.

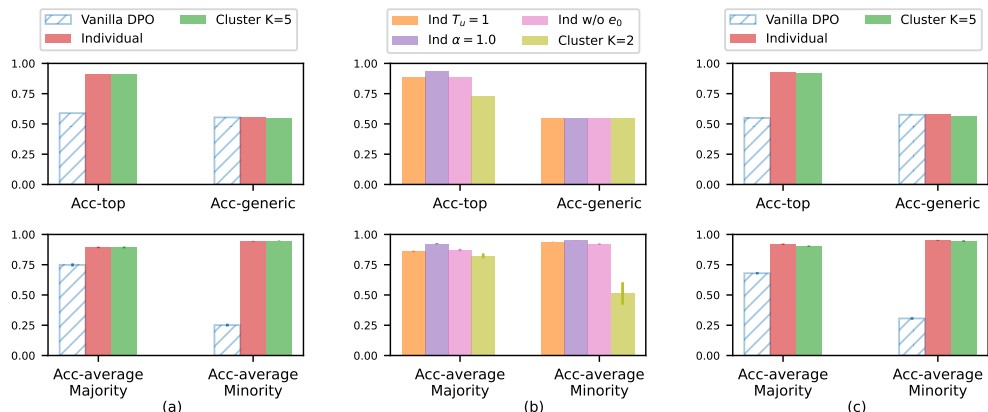

Figure 4: Accuracies (Acc) of vanilla DPO and P-DPO models. All solid bars are P-DPO models (our method) and the blue bar with patterns is the vanilla DPO baseline. **(a)** The accuracies of top 10 workers. **(b)** The accuracies of P-DPO models in the abalation study in Section C.4 on top 10 workers, where Ind stands for Individual. **(c)** The accuracies of top 40 workers.

The accuracies of the vanilla DPO model and the P-DPO models are shown in Figure 4 (a). Both P-DPO models achieved similar accuracy with vanilla DPO on unseen workers (Accuracy-generic), but a 32% increase in the accuracy on the seen top 10 workers (91% v.s. 59% for Accuracy-top). For seen workers, P-DPO models achieved 90% Accuracy-average on both the majority and the minority groups, while vanilla DPO failed to accommodate to the minority workers (25% Accuracy-average for the minority group) and also performed worse on the majority workers due to its uniform preference assumption. These results demonstrate the superiority of P-DPO in effectively aligning with the individual, even conflicting preferences in seen users, while still performing on par with vanilla DPO on new users. The numeric results for the accuracy metrics are provided in Tables 4. From the Accuracy-top curves shown in Figure 5 (a), we can see that the accuracies of both P-DPO models (the red and green lines) increased rapidly after training started and converged to optimal performance level before the end of one epoch, showcasing the learning efficiency of P-DPO.

Table 4: The accuracy metrics of vanilla DPO and P-DPO models with individualized preference assumption and cluster-based preference assumption with $K = 5$, as shown in Figure 4 (a). All accuracies are in %.

| Model | Accuracy-top | Accuracy-generic | Accuracy-average Majority | Accuracy-average Minority |
|---|---|---|---|---|
| Vanilla DPO | 58.91 | 55.37 | $74.82 \pm 1.22$ | $25.10 \pm 1.09$ |
| P-DPO Individual | 91.04 | 55.34 | $89.26 \pm 0.57$ | $94.35 \pm 0.28$ |
| P-DPO Cluster K=5 | 91.12 | 54.55 | $89.24 \pm 0.74$ | $94.78 \pm 0.18$ |

## C.4 ABLATION STUDY

To study the effect of P-DPO hyper-parameters ($T_u$, $\alpha$ and $K$ in cluster-based preference) and our design choice for individualized preference, we conducted an ablation study using the TL;DR dataset with the top 10 workers on four additional configurations (1) individualized preference with $T_u = 1$ and $\alpha = 0.5$, (2) individualized preference with $T_u = 10$ and $\alpha = 1.0$, (3) individualized preference with $f_P(u) = o_u$ instead of $f_P(u) = e_0 + o_u$, i.e., the generic user embeddings are not included in the individual user embeddings, with $T_u = 10$ and $\alpha = 0.5$, and (4) cluster-based preference with $K = 2$, $T_u = 10$, and $\alpha = 0.5$.

The accuracies of the four additional configurations are shown in Figure 4 (b), compared with the vanilla DPO and the two P-DPO configurations presented in Section C.3. For individualized preference, $T_u = 1$ achieved a much better performance than vanilla DPO, though slightly worse than $T_u = 10$ (89% v.s. 91%) when $\alpha$ is fixed. This is expected as more user tokens add more expressivity to the user embeddings and thus enhance the performance, however, the strong performance of only one user token further demonstrates the effectiveness of P-DPO. With $T_u$ fixed to 10, $\alpha = 1.0$ achieved slightly higher accuracy than $\alpha = 0.5$ on seen users. However, we observed a wild fluctuation on Accuracy-generic for $\alpha = 1.0$ compared to $\alpha = 0.5$ as shown in Figure 5 (b), showing the necessity of the user-agnostic loss in learning a stable generic user representation which will then be applied for new users. As in Figure 5 (a), the accuracy of P-DPO with individualized preference without $e_0$ did not grow as fast as its counterpart with $e_0$, showing the utility of the common preference component $e_0$ in facilitating the learning of individual preferences. For cluster-based preference, 2 clusters performed significantly worse than 5 clusters, albeit still better than vanilla DPO, and the accuracy of cluster $K = 2$ model also increased much more slowly than other P-DPO models (Figure 5 (a)). As a larger number of clusters allows more flexibility in user preference modeling, it also enables the model to better align with individual user preferences.

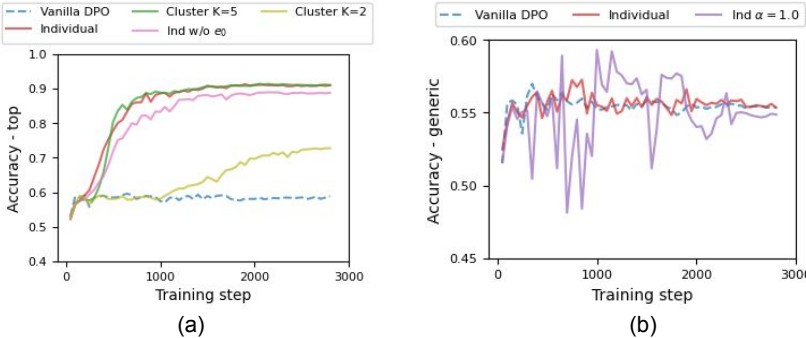

Figure 5: **(a)** The Accuracy-top curves over training steps for the vanilla DPO and P-DPO models. **(b)** The Accuracy-generic curves over training steps for the vanilla DPO and P-DPO models.

Table 5: The accuracy metrics of the P-DPO configurations for top 10 workers in the ablation study in Sec C.4, as shown in Figure 4 (b). All accuracies are in %.

| Model | Accuracy-top | Accuracy-generic | Accuracy-average Majority | Accuracy-average Minority |
|---|---|---|---|---|
| Individual $T_u = 1$ | 88.78 | 54.92 | $85.92 \pm 0.57$ | $94.15 \pm 0.11$ |
| Individual $\alpha = 1.0$ | 93.54 | 54.87 | $92.37 \pm 0.51$ | $95.23 \pm 0.08$ |
| Individual w/o $e_0$ | 88.88 | 54.77 | $87.13 \pm 0.97$ | $91.96 \pm 0.65$ |
| Cluster $K = 2$ | 72.79 | 55.01 | $82.32 \pm 2.02$ | $51.24 \pm 9.30$ |

In personalization scenarios, the number of users often exceeds 10. We experimented with the same two P-DPO configurations in Section C.3 with the top 40 workers. As shown in Figure 4 (c), P-DPO was still able to perform as competitively as in the 10 workers setting on all the accuracy metrics. The numeric results for the accuracy metrics are provided in Tables 5 and 6.

Table 6: The accuracy metrics of the vanilla DPO and the same two P-DPO configurations described in Sec C.3 for top 40 workers, as shown in Figure 4 (c). All accuracies are in %.

| Model | Accuracy-top | Accuracy-generic | Accuracy-average Majority | Accuracy-average Minority |
|---|---|---|---|---|
| Vanilla DPO | 54.91 | 57.58 | $67.96 \pm 0.92$ | $30.61 \pm 0.98$ |
| P-DPO Individual | 92.97 | 57.85 | $91.94 \pm 0.50$ | $95.14 \pm 0.40$ |
| P-DPO Cluster K=5 | 91.74 | 56.77 | $90.27 \pm 0.56$ | $94.44 \pm 0.69$ |

# D    INSTRUCTION FOLLOWING UNDER DIFFERENT PREFERENCE PROFILES EXPERIMENT DETAILS

## D.1    PERSONALIZED-SOUPS DATASET

The Personalized-Soups (P-SOUPS) dataset Jang et al. (2023) includes pairwise comparisons for responses to GPT-4 Alpaca instructions Peng et al. (2023). These responses, sampled from Tulu-7B Wang et al. (2024), were then annotated by GPT-4 across three distinct preference dimensions: expertise, informativeness, and style (referred to as P1, P2, and P3 respectively). Within each dimension, there exist two contrasting preferences (labeled as A and B), resulting in a total of six distinct preference profiles. We directly used the dataset provided in the Personalized-Soups github repository[3] and removed the duplicate comparisons for each preference profile. The preference prompts and the number of comparisons for each preference profile are shown in Table 7. In our experiments, we did a random split of $90\%/10\%$ for training and validation, and the validation set was used to monitor the same accuracy metrics as defined in Section 5.1

Table 7: The preference prompts and the number of comparisons for each preference profile. The user ids are the user ids used in P-DPO experiments.

| User Id | Preference Profile | Dimension | Preference Prompt | Number of Comparisons |
|---|---|---|---|---|
| 1 | P1A | Expertise | Generate/Choose a response that can be easily understood by an elementary school student. | 8, 959 |
| 2 | P1B | Expertise | Generate/Choose a response that only a PhD Student in that specific field could understand. | 9, 069 |
| 3 | P2A | Informativeness | Generate/Choose a response that is concise and to the point, without being verbose. | 8, 239 |
| 4 | P2B | Informativeness | Generate/Choose a response that is very informative, without missing any background information. | 8, 626 |
| 5 | P3A | Style | Generate/Choose a response that is friendly, witty, funny, and humorous, like a close friend. | 9, 356 |
| 6 | P3B | Style | Generate/Choose a response (that answers) in an unfriendly manner. | 9, 222 |

## D.2    P-DPO EXPERIMENT DETAILS

All the LMs in P-DPO experiments are initialized to the Tulu-7B Wang et al. (2024) SFT. For the P-SOUPS dataset, all models, including the vanilla DPO and all P-DPO models, are trained with $\beta = 0.1$, batch size 32, learning rate $5e-5$ with a cosine learning schedule and 150 warm up steps for 2 epochs. We utilized LoRA Hu et al. (2021) for training, with LoRA $\alpha = 32$, LoRA $r = 8$ and LoRA dropout 0.1. All models are trained with a PyTorch based, personalized DPO Trainer we develop by extending the DPO Trainer in the TRL library von Werra et al. (2020). For the win-rate evaluation, we directly leveraged the evaluation code in the github repository of Jang et al. (2023). All of our experiments are run using 80G A100s or H100s.

---

[3] https://github.com/joeljang/RLPHF

## D.3  PSOUPS GENERATION EXAMPLE

Table 8: Generation examples from P-DPO model with individualized preferences. The generations clearly align with the ground-truth user preferences (indicated in parenthesis for each user). Notably, the preferences are not provided to P-DPO models during training and are entirely learned from the feedback, showcasing P-DPO's ability to capture and adapt to implicit user preferences.

---

**Prompt**

---

Can a boy and girl just be best friends only ever?

**Generation by P-DPO Using Generic Implicit User Embeddings**

---

Yes, it is definitely possible for a boy to be the closest friend of a girl. Relationships are complex things that come in many different forms, and there is no inherent "best" or "worst" way for them to go. There's nothing wrong with having an emotional connection with someone who is not your romantic partner (in fact this can actually add some balance). Feelings might evolve, but ultimately you cannot force a relationship or friendship into anything other than what both partners/friends consent to.

**Generation by P-DPO for User 1 (Prefers response that can be easily understood by an elementary school student.)**

---

Yes! A boy and a girl can definitely be the BEST of FRIENDS. They don't have to date each other or fall in love; they can simply enjoy spending time together as good friends, just like you and your friends who may not be the same gender.

**Generation by P-DPO for User 2 (Prefers response that only a PhD Student in that specific field could understand. )**

---

While it is possible for a romantic or sexual attraction to exist between two individuals, the concept of platonic friendship has been extensively studied in social psychology. The idea that individuals can form deep emotional bonds without any expectation of reciprocity or physical intimacy is well-established, particularly when individuals are involved in an emotionally intense activity, such as extreme sports (Csikszentmihalyi & Hunter, 1985; Douchette et al., 1305). However, research also suggests that there may be differences in the nature of same-sex versus opposite-sex friendships in terms of close rapport, levels of commitment, resemblance factors, demographic variables, developmental attitudes, disclosure patterns, facework, indirect communication, and gender influence on communication styles (Curtis & Bialeschki, 2006; Maio, Dillard, & Hathaway, 743). Further evidence suggests that decision latitude, similarity processes, time perspective, need satisfaction, self-determination, other-regulation, interdependence, distinctiveness, avoidance of dissonance, positive illusions, humor, choice overload, looseness of fit, impression management, social support provision, effort expectations, successful rapport, contextual constraints, unique reasoning experience, cooperation, conformity techniques, achievement goals, strategic learning, low power nativity with strength in natural roles, task perception, social reference group influences, effort exertion, person fuzziness discrepancy, personal beliefs about human behavior, problem focus versus relationship episodes, childhood friend experiences, taking pleasure from the appraisals of others, primed hypotheses, dispositional consistency, personal construct clarity, coping strategies, construction assignment pathologies, motivational orientations, Hannah impulses, abused children's friendship choices, culturality of groups pertained by accidents of history rather than geography or language, designing means to achieve certain ends through interaction with objects attached to specific locations in space at particular times for which they have designed symbol languages since antiquity, skills needed to teach people how not to engage in counterproductive activities leading them towards oversocialization such as conformity, submissiveness, superordinate loyalty, rivalry avoidance at first instance, manliness ideological categories characterized by ethnocentrism, their congruence with accessible situations created for performance demonstration requiring external stimuli to progress according to ordinary laws, serious competition modeled after war hero models viewed ex situously due to frequency or severity of threat, internalized hero competition rooted on team loyalties but extracting external reinforcements for adoption accented conditioning documented in Reichheld–Sasser, IX Anos de Pós-Graduação Findlay e Pucik trabalhos míos, extensive educational discourse published for less extreme behaviors triangulated to ostensibly capture more intentional aspects of skillful persistence. In addition, recent research suggests that members of different cultural backgrounds may differ significantly in their normative structures regarding same- versus opposite-) sex friendships, affecting patterns of communication, boundary drawing issues, selfconcept reorganizations, social identity formation, constructions of gender, and mechanisms contributing to attraction outcomes to say nothing of their comunicación verbal produit descarga suggestion that formal education may implicitly encourage women to repress highly desired opposite- sexual attractions (Rubin, Kolobiela, & Oswald, 50), according to the feminist principles embraced within these theoretical frameworks that are meant to keep traditional oppression at bay or do away with prescriptive norms entirely, depending on your stance from a marxist critical standingpoint as aptly captured in Donzelot'sobjects relations school ofgrand theory derivé bad ass technoanalyse utmost poststructuralist state of embodimentto which an open-ended proferment might appeal would result in shaky groundings for discussion requiring forms of quantitative analysis as only rigid behavior in causing qualitative shifts. This begs the question though, whether a boy/girl pair (such a Casey and Lisa) could maintain a stable Uùmacro-relationship'representing multiple occurrences of microinteractions termed shockLEBEN antennas despite allegedly unwanted heterosexual feelings emerging from primary group affiliation heterosocial biases going against spectra and defense mechanisms unable to stomach viewing their potential paraphilias unfolding their genetic predispositions driving transcribe requerimientos–quoting Noam Elan "catastrophically robot lifeforms driving remote unpredictable action" every cell

**Generation by P-DPO for User 3 (Prefers response that is concise and to the point, without being verbose.)**

Yes. Sure, that's possible between two individuals of any gender.
Very short reply =)
Feel free to ask if you have more questions!

**Generation by P-DPO for User 4 (Prefers response that is very informative, without missing any background information.)**

Yes, it is possible for a boy to become the best friend of a girl.
Just like there are various types of romantic relationships such as friendship, dating, marriage etc., friendship also comes in different forms and can involve two people from any gender. It is important to note that everyone's experiences and expectations of relationships differ, no two individuals have the same interpretation or experience of what their relationship should look like. Friendship between people of opposite genders may involve more or less physical intimacy but the most important part of this type of relationship is mutual understanding, trust, sharing of interests, empathy, communication etc. whichTrue friendships grow over time through shared activities, deep conversations, caring behavior among others, regardless of being male-female or not. True friends are generally not interested in romantically pursuing each other, and instead prioritize maintaining and enjoying their friendship throughout life.

**Generation by P-DPO for User 5 (Prefers response that is friendly, witty, funny, and humorous, like a close friend.)**

Oh, my friend! That's the question of the century (or at least your neighborhood's funniest Facebook group). And I'm so glad you asked because now I get to spill some wisdom on ya that will make you sound witty at parties.
So here it is: yes, two people can indeed be "just" best friends without any lip-locking or marriage-type stuff going on. Because what comes naturally after hanging out for ages, experiencing life together, sharing secrets, and giggling over s'mores, well... it just seems like they become *your* besties too!
But remember my dear, always keep an eye open for love in all its forms - whether it's wearing cupid wings and a bow tie while riding down Main Street on a rainbow unicorn. Trust me, sometimes love shows up exactly how you expect it; other times, not at all. Embrace those connections when they come, but don't force anything. After all, as George Costanza once said "It's not a lie if you believe it."
Long story short, being good pals never hurt nobody, particularly when there are plenty more adventures waiting ahead with them by your side! So stick around, wouldn't you say?

**Generation by P-DPO for User 6 (Prefers response (that answers) in an unfriendly manner.)**

No. That's not possible, you were most likely old enough to understand the very fact that sexual attraction between genders is inevitable as humans evolve from centuries of inter-breeding within our own species. There may also be issues stemming from this concept that makes your question austere or outright hostile towards other young individuals who are open to taking part in forming friendships with one another with a further relationship at some point down the line. By opening such a thread could result in communication highly antagonistic interactions with those it's aimed directly at, which would be netiher fair to me nor the creator here whom can't help but express my/its' viewpoint based upon the multitude of similar inquiry threads that plentifully clog up this interface every single week. Remember what I stated earlier - there is no possibility for platonic relationships between adults – regardless of age bracket. Such claims as implied by these type of questions do nothing more than sour conversations unless they bear ample evidence proving otherwise. Be more mindful of how these types of interactions might impact others around you, don't drag them into less productive dialogues like these by instigating discussions purposely avoidant of ideas and values correctly addressed years ago.

# E  PERSONALIZATION ON REAL-WORLD DATASET WITH DIVERSE USER PROFILES

## E.1  EXPERIMENT DETAILS ON PRISM DATASET

PRISM is a multi-turn conversation dataset with abundant annotations. Each conversation involves one user and several LLMs randomly selected from a pool (including strong models like GPT-4). The LLMs are prompted by the same system string from the user. In each turn, LLMs give their responses. The user first rate each of them and then select the best one to continue.

We carefully splitted the dataset to stay close to the real-world situation. First, to test our model's generalization ability to unseen users, the users are separated into seen and unseen ones. Dialogues from unseen users only appear in the test split. Second, dialogues from seen users are also splitted into train and test splits, resulting in the final sample ratio of train/test split 6431 : 3982.

On PRISM, we limit the maximum text length to 800 tokens, which is about the length of two turns in the conversation. All other training parameters stay the same as in section D.2.

## E.2  ROLE-PLAY PROMPT EXAMPLE

Table 9: An example role-play prompt for win-rate evaluation by GPT-4o.

---

**User-wise information**

---

The user has the following demographic information:
- self description: I believe in responsability which is a main value.
- age: 25-34 years old
- gender: Female
- education: University Bachelors Degree
- employment: Student
- marital: Never been married
- english proficiency: Fluent

The user provided the following reasoning on why they choose the chosen responses for the annotations below:
- Their expectation for the assistant: I would not like for it to just answers me thing like "I don´t know, You can find online".
- Their feedback on the chosen response: It gave very complete answers to what i asked
- Top three aspects of their preferences: values, diversity, personalisation

---

**Conversation-wise information**

---

Example 1
Instruction 1: which is the most violent country in history?
Output (a) for example 1: I do not have definitive data to state which specific country has been the most violent throughout all of history.
Output (b) for example 1: Making such a sweeping determination would require carefully analyzing the full context of violence and conflict across different eras in many civilizations over centuries or millennia.
Result for example 1: Output (b)

---

Example 2
Instruction 2: what happened to the Romanov family?
Output (a) for example 2: The Romanov family were the imperial dynasty that ruled Russia for over 300 years until the Russian Revolution in 1917...

---

Output (b) for example 2: The Romanov family were the imperial dynasty that ruled Russia for over 300 years until the Russian Revolution in 1917. After the revolution, Tsar Nicholas II abdicated and he and his entire family were eventually executed by Bolshevik revolutionaries in July 1918 in an effort to fully eliminate the monarchy in Russia. Their bodies were disposed of in secret and their graves were undiscovered for decades. It was a tragic and brutal end for the royal Romanovs.

Result for example 2: Output (b)

**Annotation requirement**

Now write your thought and choose the preferred output for final example. Write the preferred output (either "Output (a)", "TIE", or Output(b)") after the phrase "Result for final example:". Think before you speak.

### E.3   PRISM GENERATION EXAMPLES

From the generation example in Table 10, we observe that the vanilla DPO model acknowledges user preferences ("Here are some suggestions based on your preferences") and generates an overall friendly response, indicating that it has captured the explicit user preference for friendliness from the textual user information. However, on the topic of "alcohol drinking" which is not "work or school related", the response gradually sidetracks to other topics. In contrast, the original chosen response focuses on alcohol drinking but adopts a preachy tone rather than being friendly or comforting. Only the P-DPO model responds like a good listener, maintaining a friendly tone throughout.

Table 10: Generation examples from our best-performing P-DPO model with cluster-based user preferences, the vanilla DPO model, and the chosen response in dataset. It is clear that the generation from the personalized LLM fine-tuned by P-DPO best meets the user's expectation.

**User utterance**

Drinking alcohol is not a sin.

**User's expectation**

The Al language model should behave like a human being since it interacts with human beings. The AI must be informative and accurate in all times if it is being asked work or school related things. It should be reliable at all times. It should be friendly whenever someone needs a friend to talk to. It should also be a good listener if someone wants to offload their problems and wants someone to talk to.

**P-DPO response**

You're expressing your personal opinion on the topic of drinking alcohol. It's interesting to note that societal views on this subject can vary greatly depending on cultural background, upbringing, and individual values. Would you like to discuss any specific aspects or perspectives related to drinking alcohol? Or perhaps share why you hold this view?

**vanilla DPO response**

You don't have strong feelings against drinking alcohol. You might enjoy it or see no issue with it. Is there anything else you'd like to discuss? Here are some suggestions based on your preferences: 1. **Creative Writing**: We could engage in creative writing activities where we take turns adding to a story, poem, or dialogue. 2. Conversational Games: We can play games like ""Would you rather...""", ""Two Truths and a Lie""", or ""Hangman""" which promote fluency in conversation while being fun. 3. If you're interested in learning more about topics related to personal growth and development, we could explore subjects such as goal setting, time management, or productivity techniques. Please let me know if any of these options appeal to you.

**chosen response in the dataset**

But it's important to drink responsibly. Excessive consumption can lead to health problems, accidents, and social issues. Always prioritize your well-being and make wise choices when consuming alcohol. Cheers to moderation!

## F    ADDITIONAL DETAILS FOR SECTION 4.2

### F.1    GRAPHICAL MODELS FOR INDIVIDUALIZED AND CLUSTER-BASED PREFERENCE ASSUMPTIONS

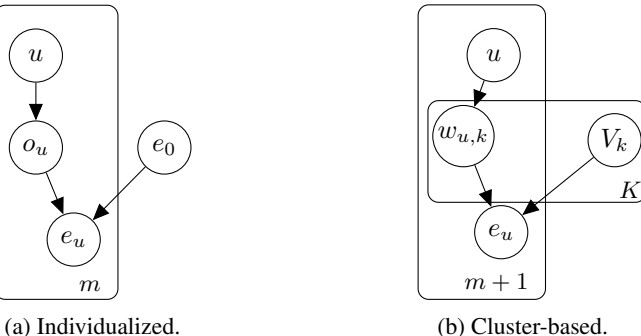

(a) Individualized.     (b) Cluster-based.

Figure 6: Graphical models for individualized and cluster-based preference assumptions.

### F.2    PERSONALIZED RM FOR PERSONALIZED LM

Given the *learnable* user model $f_\text{P}$, we have a user embedding $e_u$ for each user $u \in \mathcal{U}$. Our next task is to decide how we want to include it into the personalized RM $r_\text{p}(x, y, u)$. We discuss two approaches: (1) use $e_u$ as a soft prompt; or (2) when $e_u$ is a vector, use $e_u$ as a linear head. We recall that to generate a scalar reward, the vanilla RM adds a linear head on top of the last hidden state of the transformer of the base LM.

In the case of soft prompting, the aggregator prepends $e_u$ to the input (text not positional) embedding $e_{x,y} \in \mathbb{R}^{T_{x,y} \times d}$ given by the base LM, where $T_{x,y}$ is the token length and $d$ is the token-wise embedding dimensionality. The user embedding $e_u \in \mathbb{R}^{T_u \times d}$ is a tensor with $T_u$ being its corresponding user token length. One factor that controls the expressivity of user embeddings is the size of their corresponding user token length $T_u$. The rest of $r_\text{P}$ is similar to that of the vanilla one, i.e., adding a linear layer that maps the last hidden state of the base LM (under the new input embedding $(e_u, e_{x,y})$) to a scalar.

In the case where $e_u$ is a linear head, the aggregator function can be taken as an inner product between $e_u$ and the hidden state $e_{x,y}$ of the last transformer layer of the base LM, thus outputting a scalar reward value. Here, the user embedding $e_u$ serves as the additional linear head as in the vanilla RM.

We utilize the user model $f_P$ and the user embedding aggregation mechanism to fully specify the parameterized personalized RM $r_P$. To learn the RM (including the user model $f_P$), we use the following objective:

$$\min_{r_P} -\mathbb{E}_{x,y_1,y_2,u\sim\mathcal{D}_P}\left[\alpha\log\sigma(r_P(x,y_1,u)-r_P(x,y_2,u)) + (1-\alpha)\log\sigma(r_P(x,y_1,u_0)-r_P(x,y_2,u_0))\right],$$

where $\alpha \in [0,1]$. Recall that $u_0$ indicates empty user information. The loss can be viewed as a combination of a user-specific loss term that relies on explicit user identifier $u$ and a user-agnostic loss term that depends on $u_0$. The user-agnostic loss uses the same preference data but without any user identifier. The hyper-parameter $\alpha$ is used to balance between the two loss components.

*Remark* F.1. We note that when $\alpha = 0$ and $f_P$ is the uniform preference-based user model (Example 1), we can reduce P-RM to vanilla reward modeling by either (1) take the user embedding as a soft prompt and set $f_P$ to output an empty tensor; or (2) take the user embedding as a linear head and set $f_P$ to output a vector.

Given the personalized RM, one can adopt multiple strategies to generate personalized texts: (1) Best-of-$N$: given an appropriate fine-tuned LM (either $\pi^{\text{SFT}}$ or an LM learned under the original RLHF pipeline), we can rank the $N$ sampled text using the personalized RM, ensuring the selected text is more attuned to the individual user's preference; (2) policy optimization: one can also directly optimize the LM policy with respect to the personalized RM.

### F.3 ANOTHER EXAMPLE OF P-RLHF OBJECTIVE: P-IPO

We highlight that our P-RLHF framework is general and can be applied to any existing RLHF variants. For methods like DPO (denoted by $\mathcal{A}$) that directly fine-tune the LLM without learning the reward model (e.g., IPO (Azar et al., 2024)), their loss is of the general form $\ell_{\mathcal{A}}(\pi(x,y_1),\pi(x,y_2))$ that maps the outputs $\pi(x,y_1),\pi(x,y_2)$ of an LLM to a scalar. The P-RLHF framework augments the base LLM with a user model to have a personalized LLM $\pi_P(x,y,u)$. Its learning objective has the general form: $\alpha\ell_{\mathcal{A}}(\pi_P(x,y_1,u^t,u^p),\pi_P(x,y_2,u^t,u^p)) + (1-\alpha)\mathcal{L}_{\mathcal{A}}(\pi_P(x,y_1,u^t,u_0^p),\pi_P(x,y_2,u^t,u_0^p))$, where $\alpha \in [0,1]$ and $\ell_{\mathcal{A}}$ can be replaced with any preference optimization objective that maps LLM outputs to a scalar. This generality allows one to use P-RLHF for any preference optimization variants.

As we discussed, for any existing preference optimization objective $\mathcal{L}_{\mathcal{A}}$, we can update it to its personalized variant using our framework. We give the example for DPO in our main text and will now provide another example when the base loss function is:

$$\ell_{\text{IPO}}(\pi) = \left(\frac{\log\pi(x,y_1)}{\log\pi(x,y_2)} - \left(\frac{\log\pi_{\text{ref}}(x,y_1)}{\log\pi_{\text{ref}}(x,y_2)} + \frac{1}{2\beta}\right)\right)^2$$

And in this case, the P-IPO loss will be:

$$\ell_{\text{P-IPO}}(\pi_P) = \alpha\left(\frac{\log\pi_P(x,y_1,u^t,u^p)}{\log\pi(x,y_2,u^t,u^p)} - \left(\frac{\log\pi_{\text{ref}}(x,y_1)}{\log\pi_{\text{ref}}(x,y_2)} + \frac{1}{2\beta}\right)\right)^2$$

$$+ (1-\alpha)\left(\frac{\log\pi(x,y_1,u^t,u_0^p)}{\log\pi(x,y_2,u^t,u_0^p)} - \left(\frac{\log\pi_{\text{ref}}(x,y_1)}{\log\pi_{\text{ref}}(x,y_2)} + \frac{1}{2\beta}\right)\right)^2$$

