# OpenReview forum: "Personalized Language Modeling from Personalized Human Feedback"
_ICLR.cc/2025/Conference — Submitted to ICLR 2025_

### Official Review · Reviewer_hQjp · 2024-10-31

**Soundness:** 3
**Presentation:** 2
**Contribution:** 2
**Rating:** 3
**Confidence:** 4

**Summary:**

This paper introduces a framework called Personalized Reinforcement Learning from Human Feedback (P-RLHF) aimed at personalizing large language models (LLMs) based on individual user preferences. The authors propose a lightweight user model that jointly learns user preferences (both explicit and implicit) and the LLM. The key claims are that this framework scales efficiently with increasing users, handles diverse feedback, and reduces the need for explicit user prompts.

**Strengths:**

- The paper addresses an important area of research—personalizing LLMs based on user feedback.
- The proposal to use both explicit and implicit feedback for personalization is valuable and well-motivated.
- The experiments cover several datasets, which helps validate the general applicability of the approach.

**Weaknesses:**

- The paper does not sufficiently address how implicit feedback is effectively used or how it handles contradictory user preferences.
- The scalability claims, while central to the paper, are not backed up by detailed computational analysis or resource cost comparisons with other approaches.
- There is limited discussion on how this approach performs in real-world dynamic environments where user preferences might shift over time.
- The experimental results, while positive, are not sufficiently compelling to clearly demonstrate the advantage of P-RLHF over simpler personalization methods.

**Questions:**

1. How does the model ensure that implicit feedback does not conflict with explicit user preferences, and how does it resolve such conflicts?
2. Can the authors provide more detailed information on the computational efficiency of P-RLHF? How does it compare in terms of time and memory consumption to other RLHF approaches?
3. How does the framework handle dynamic user preferences that may change over time, and how quickly can it adapt?

---

> ### Author Response · Authors · 2024-11-29
>
> We thank the reviewer for their feedback. We appreciate that they found our research topic important, our method valuable and well-motivated, and our experiment results positive and generally applicable. Below we provide responses to the weaknesses and questions:
>
> **Weakness 1 & Question 1**: Contradictory user preferences
>
> We want to highlight that in the paper, when we mention “conflicting preferences,” we meant that in **a population of users**, there could be different people who have conflicting preferences. This is **not** saying that for a single user, the user has conflicting preferences in their explicitly written preference and their provided preference data/label. The second case happens when the user says that they prefer one type of response but when they label the preference data, they always label the response that does not belong to that type to be preferred.
>
> In the first case where there is a population of users who have different and possibly conflicting preferences, as shown in our experimental results (Section 5.1, 5.2, 5.3), our method can learn them and personalize model responses based on that. That is, for user A who prefers polite responses and user B who prefers impolite responses, P-RLHF can learn and adjust to their individual preferences.
>
> Moving on to the second case, let’s consider the setting where a user explicitly specified that they want “polite response” but has labeled the impolite response to be preferred over the polite one in all their preference data. In this case, the model learning objective has two parts in tension because on one hand, the objective has the preference optimization part to encourage the learned model to best fit the preference data (e.g., put more probability on impolite responses over polite ones in this case). But on the other hand, the objective also has the term specifying the constraint that the KL divergence between the learned model and the SFT should not be too large. That is, conditioned upon the user provided context “I prefer polite responses”, the learned model $\pi(\cdot | \text{“I prefer polite responses.”, prompt} )$ should not deviate much from $\pi_\text{SFT}(\cdot | \text{“I prefer polite responses.”, prompt} )$. Because of this tension in the learning objective, the end behavior of the learned model will depend on which part has higher values. That is, it will depend on the size of the KL divergence penalizer $\beta$ as well as the exact log-probability on the chosen and rejected responses for the LLM. In this case, what the implicit user embedding learns will depend on the aforementioned values. For example, in cases where the two parts have relatively similar values, then the implicit user embedding will play the role to balance off the explicit user embedding so that the learned model will output polite and impolite responses around half and half.
>
> **Weakness 2 & Question 2:** Efficiency/Scalability claims
>
> We thank the reviewer for asking this. To be more clear, in training P-RLHF models, the total number of trainable parameters (#Total) = # of trainable parameters for the LLM (#LLM) + # of trainable parameters for the user model (#User). When we say “lightweight”, we meant that #User $\ll$ #LLM. For example, for our K-cluster user model #User = (# of users x K) + (K x # user tokens x input_embed_dim) where input_embed_dim is the input embedding dimension for the LLM. When we are training Llama-3 8B cluster model, #User is actually **10x smaller** than #LLM.
>
> This is in contrast to other existing RLHF methods for building personalized LLMs (e.g., [1][2]), since most of them require training multiple LLMs. That is, their #Total = #LLM * N where N >=2 and oftentimes, N can be quite high.
>
> As such, our method is lightweight in both memory and computation, as our trainable parameters are much smaller than the LLM trainable parameters and other RLHF personalization methods’ trainable parameters. We have added these discussions in Section 5 of the revised draft.
>
> [1] Jang, Joel, et al. "Personalized soups: Personalized large language model alignment via post-hoc parameter merging." arXiv preprint arXiv:2310.11564 (2023).
>
> [2] Wang, Kaiwen, et al. "Conditional Language Policy: A General Framework for Steerable Multi-Objective Finetuning." arXiv preprint arXiv:2407.15762 (2024).

---

> ### Author Response · Authors · 2024-11-29
>
> **Weakness 3 & Question 3:** Dynamic environment and shifting user preferences
>
> Our methods **naturally work with dynamic environments** where users put in their prompts and interact with the model dynamically. P-RLHF allows for adding real-time user textual information (their chat history, additional new information of the users, etc.) to generate personalized responses. In fact, as shown in our PRISM experiment (Section 5.3), our method works well with changing textual information of the users as they interact with the chatbot. If one were to update the implicit user embedding as they collect more data, they can retrain the model. Due to the modular nature of our design, instead of needing to fine-tune the LLM, they can just fine-tune the user model. In particular, for cluster-based user models, we can just train the weight vector for users they want to update their embeddings for while holding other parameters frozen. As we discussed in our response for W2 and Q2, the number of trainable parameters in this case is much smaller than the trainable parameters than the language model itself.
>
>
> **Weakness 4:** Significance of experimental results
>
> In all three experiments, our method consistently outperforms strong baselines, which leverage either ground-truth preferences directly (Section 5.2) or user-provided preferences and information (Section 5.3). Our experiments cover a range of scales, from fully controlled simulated user preferences (Section 5.1) to **realistic scenarios involving nuanced, multi-faceted user preferences, with 1,500 users and 8,000 conversations in the PRISM dataset** (Section 5.3). Notably, to the best of our knowledge, our work is the first to conduct experiments with PRISM—**the largest publicly available real-world personalized preference dataset** featuring diverse user profiles and conversation topics. The effectiveness of our method is demonstrated on this large-scale dataset by surpassing strong baselines (Table 2, also in our new experimental results in the general rebuttal) and enabling the fine-tuned personalized LLM to generate high-quality responses tailored to individual user preferences (Appendix E.3). Therefore, we believe **our experiment results clearly establish the effectiveness of P-RLHF and underscore the significance of our contribution**.
>
> We would greatly appreciate it if the reviewer could elaborate on what aspects are lacking in our experiments. This feedback would help us refine our work and better demonstrate the effectiveness of P-RLHF in a more compelling way.

---

> ### Author Response · Authors · 2024-12-03
> **Last day for response: Follow up on our rebuttal**
>
> Dear Reviewer hQjp,
>
> As the discussion period is coming to a close, we would greatly appreciate it if you could let us know whether our rebuttal has fully addressed your concerns and questions.
>
> We have provided point-by-point responses to your concerns and questions, including:
>
> - We have clarified the setting of conflicting preferences in our experiments and demonstrated how our method effectively handles such conflicting preferences. Additionally, we provided further analysis on performance under other conflicting preference scenarios, illustrated with a qualitative example.
>
> - We added a paragraph in the revised manuscript analyzing the scalability of our method, specifically highlighting how the number of trainable parameters is significantly smaller compared to other RLHF-based personalization methods.
>
> - We have elaborated on how our method can handle shifting user preferences in a dynamic environment through either explicit user textual information or small updates of user models.
>
> - The significance of our experimental results is acknowledged by other reviewers:
>     - Reviewer oVq3: “In both synthetic and human evaluations, P-DPO significantly outperforms DPO”;
>     - Reviewer rjtj: “...showing that the personalized model excels in understanding and responding to user needs, validating its effectiveness in practical applications”;
>     - Reviewer uiaG: “The personalized LLMs showcased in the study hold significant practical value, and the paper successfully achieves personalized content generation on the utilized datasets. This effectively highlights the framework’s capability to cater to diverse user preferences in real-world applications.”
>
>     We emphasize that our method is evaluated on the **largest publicly available real-world personalized preference dataset**, and the win-rate gain of our method over strong baselines is high (over 15%-20% when evaluated with 70 users, 256 conversation turns). We would greatly appreciate your further elaborations on how we could make our results even more compelling.
>
> Please let us know if there are any additional analyses or results we could provide to further address your concerns. Thank you for your time and consideration.
>
> Best regards, \
> Authors of Submission 12971

---

### Official Review · Reviewer_uiaG · 2024-11-01

**Soundness:** 2
**Presentation:** 3
**Contribution:** 3
**Rating:** 6
**Confidence:** 4

**Summary:**

This paper introduces Personalized Reinforcement Learning from Human Feedback (P-RLHF), a novel framework designed to enhance the personalization capabilities of large language models (LLMs) by integrating both explicit and implicit user models. By leveraging user-specific information and feedback, P-RLHF effectively captures diverse and individualized user preferences, addressing the limitations of traditional RLHF methods that assume uniform user preferences.
The authors validate the efficacy of the P-RLHF framework through a series of comprehensive experiments, ranging from controlled synthetic settings to large-scale real-world datasets. These experiments demonstrate that P-RLHF not only successfully captures and adheres to individual user preferences but also maintains high performance and scalability in diverse application scenarios. The clear and logical presentation of the framework, coupled with intuitive response generation results, underscores the practical value and potential impact of personalized LLMs in real-world applications.
Overall, this paper makes substantial contributions to the field of personalized LLM by presenting a well-structured framework and a novel optimization method that together advances the state-of-the-art in personalized language model training.

**Strengths:**

1. The paper presents the P-RLHF framework, which integrates both explicit and implicit user models to facilitate personalized learning from user feedback, thereby introducing a novel approach to the research on personalized LLMs. The introduction of P-DPO represents an innovative optimization method that effectively balances personalization and generalization for both known and unknown users. This approach demonstrates excellent scalability while addressing the personalized needs of multiple users, marking a significant advancement over traditional RLHF methods.
2. The overall structure of the paper is exceptionally clear, methodically introducing the background, identifying the problem, proposing solutions, and validating the experiments related to personalized RLHF. The logical progression facilitates easy comprehension for readers. Each experimental section is meticulously designed with clearly defined objectives, enabling readers to effortlessly track the innovative aspects of the methodology and their corresponding outcomes.
3. The response generation results provided in the paper offer a clear and intuitive demonstration of how different user preferences are handled in a personalized manner. The personalized LLMs showcased in the study hold significant practical value, and the paper successfully achieves personalized content generation on the utilized datasets. This effectively highlights the framework’s capability to cater to diverse user preferences in real-world applications.

**Weaknesses:**

1. Zero-Length Responses in Experiment One: The occurrence of zero-length responses, which the authors justify as an expected outcome via mathematical proofs, raises questions about whether an LLM should indeed produce responses with no content. This result appears to stem from a polarized experimental design and methodology, which may not reflect practical application scenarios or user expectations. A more nuanced experimental setup could better balance realistic use cases with the need for distinct preference modeling.
2. Figure 1 Lacks Identification of Key Modules: While Figure 1 presents the overall framework, it does not clearly indicate critical components introduced by the authors, such as P-DPO. And there is no explanation of Figure 1 in the main text. This omission complicates the interpretation of the framework diagram, hindering the reader’s ability to intuitively grasp the structure and contributions of the proposed approach.
3. Insufficient Ablation Study: The ablation experiments, which appear only in the appendix, focus solely on parameter variations rather than examining which specific design choices contribute to the improved performance over baseline models. For example, it is unclear whether the personalized capabilities remain robust when only the implicit model is used without the explicit component. The paper lacks a thorough investigation into such aspects, leaving questions about the impact of each module unanswered.
4. Binary Nature of Evaluation Metrics: Both the preference modeling and experimental evaluation adopt a highly binary approach. In modeling preferences, options are presented as direct opposites (e.g., long vs. short responses), which likely contributes to the appearance of zero-length responses. Furthermore, evaluation is limited to win-rates relative to baseline models, which does not provide a clear view of the actual performance gains. Since the goal of personalization is to improve user satisfaction, more nuanced analyses, rather than a strictly binary approach to the generated content, are necessary to substantiate the effectiveness of the approach.
5. Lack of Consideration for Contextual Shifts in User Preferences: User preferences can be context-dependent, particularly when involving multidimensional combinations of preferences. In specific scenarios, user preferences may shift according to immediate needs or context. The reliance on a static dataset overlooks these potential dynamics, and the paper does not address how such variations might impact model performance.

**Questions:**

1. An explanation is requested for the occurrence of zero-length responses presented in Section 5.2 and Figure 3. Logically, even for users who prefer concise replies, it is expected that responses maintain a reasonable lower bound on length rather than resulting in either no content or excessively long responses. The current approach appears overly polarized and does not account for practical application scenarios where balanced response lengths are desired.
2. It is recommended that Figure 1, which outlines the proposed module design, be incorporated directly into the main body of the text. This integration would enhance the comprehensibility of the framework by ensuring that the figure is closely tied to the corresponding narrative, thereby making the overall framework easier to understand for readers.
3. I suggest the author add more nuanced analyses, rather than a strictly binary approach to the generated content, which are necessary to substantiate the effectiveness of the approach.

---

> ### Author Response · Authors · 2024-11-29
>
> We thank the reviewer for their feedback. We are encouraged that they found our presentation exceptionally clear, recognized the excellent scalability of our approach, and acknowledged its significant advancement over traditional RLHF and strong practical value. Below we provide responses to the weaknesses and questions:
>
>
> **Weakness 1 & Question 1**: Zero-Length Responses in Experiment One
>
> We emphasize that this experiment is specifically designed to demonstrate that personalized LLMs trained using our method can achieve the expected optimal behavior in a fully controlled setting. Under such a setting, the expected optimal behavior—zero-length generation for minority workers who consistently prefer shorter responses—can be **rigorously derived** (Section 5.1, Expected behavior of the optimal personalized LLM paragraph, E1). As shown in the results, our method effectively achieves this expected behavior, demonstrating its capability to align LLMs with different individual user preferences.
>
> Building on this proof-of-concept, we further validate the effectiveness of our method for realistic use cases through two additional sets of experiments: **Instruction Following under Different Preference Profiles** (Section 5.2) and **Personalization on Real-World Preference Dataset with Large User Base** (Section 5.3). In Section 5.2, our method achieves win-rates > 50% against a strong baseline (SFT model prompted with ground-truth preference profiles), and in Section 5.3, it similarly outperforms a strong vanilla DPO baseline which directly leverages user-stated preferences and sociodemographics.
>
> Moreover, in these two experiments, **the personalized LLMs trained with our method generate coherent responses tailored to individual user preferences, as evidenced by the generation examples provided in Appendix D.3 and E.3**. These quantitative results and qualitative generation examples collectively demonstrate that our method is capable of learning personalized LLMs which produce high-quality, personalized responses in realistic settings with diverse user preferences.
>
>
> **Weakness 2 & Question 2:** Figure 1 Lacks Identification of Key Modules
>
> Thank you for your suggestions and recommendations. To address the concern and improve readability, we have made the following revisions in the updated pdf:
>
> - Modified Figure 1 to better highlight the key components of our P-RLHF framework, namely the **user model**, which maps user information to **user embeddings** (both explicit and implicit), and the **base LLM**, which is trained jointly with the user model using the P-DPO learning objective. Additionally, we revised Figure 1 caption to emphasize these key modules and included references to the corresponding sections for further details about each component.
> - Added references to Figure 1 in both Section 1 (Introduction) and Section 4 (Methodology) to ensure that Figure 1 is explicitly connected to the framework discussion in the main text.
>
> All our edits have been highlighted in red for ease of reference.

---

> > ### Comment · Reviewer_uiaG · 2024-12-02
> >
> > Thank you for your comprehensive and detailed response.
> > 1. I can understand that Experiment 1 represents an extreme condition, and the authors explained that more realistic application scenarios are covered in other experiments. However, given that Experiment 1 also employs the proposed method, it is still somewhat puzzling that the same method could produce such extreme and unreasonable results, which the authors should have anticipated.
> > 2. I am pleased to see the authors’ modifications to Figure 1. Although the names of the proposed modules are still not explicitly included in the figure, they have been highlighted. And combined with the caption, the understanding is relatively clear.
> > 3. I am glad to see the authors’ addition of ablation studies and datasets, which make the experimental results more comprehensive than before.
> > 4. I appreciate that the authors acknowledged the limitations of their previous experimental settings and supplemented more complex and realistic experiments to overcome these limitations, making the results more convincing.
> > 5. The authors have added targeted explanations to clarify the model's adaptability under the proposed conditions, which indeed helps to alleviate readers' concerns in this regard.
> >
> > Overall, the authors have shown great attention to the reviewers’ comments and made proactive responses and additions, which is commendable, although these improvements should ideally have been made in the initial submission. Considering this, I am willing to raise my score to 6.

---

> > > ### Author Response · Authors · 2024-12-03
> > > **Thank you for your response**
> > >
> > > Dear Reviewer uiaG,
> > >
> > > Thank you for your detailed response! We are glad to see that you found our rebuttal and additional experiments helpful.
> > >
> > > Regarding point 1, we would like to elaborate a bit further on Experiment 1. This experiment is synthesized in a fully controlled setting with a very simple preference dimension—response length. Under our personalized DPO learning objective, the LLM quickly and effectively learns individual user preferences and achieves the expected optimal behavior—generating zero-length responses for minority users who always prefer shorter responses. We hypothesize that the model was able to learn this behavior because the ground-truth reward is very simple in this case (thus easy to be learned) compared to much more complex underlying rewards in real-world settings (Experiment 2 and 3).
> > >
> > > We highlight that this experiment serves only as a proof-of-concept to verify that our method can learn LLMs which demonstrate ideal behavior, and more importantly, it allows us to perform a more in-depth analysis of the learned LLMs beyond standard metrics such as win-rates.
> > >
> > > Thank you again for your feedback!
> > >
> > > Best regards, \
> > > Authors of submission 12971

---

> ### Author Response · Authors · 2024-11-29
>
> **Weakness 3:** Insufficient Ablation Study
>
> For the design choices, we have experiment results on whether it is helpful to have the implicit user embeddings in the PRISM experiments (Section 5.3) . Our results in Table 2 have shown the benefits of adding implicit user embeddings in addition to explicit user textual preferences—win-rate > 50% when compared to vanilla DPO with only explicit user textual information.
>
> Per the reviewer’s suggestion (and we thank the reviewer for this suggestion), we have added an additional ablation study for the PRISM experiment in the general rebuttal. We added an experiment which only includes the implicit user model in the personalized LLM. From the results shown in the general rebuttal, we can see that P-DPO with only implicit user model outperforms the same two baselines by achieving > 50% win-rates, when evaluated on all conversation turns from 20 or 70 randomly sampled users. These new experiment results demonstrate that in a real-world dataset with large user bases, the personalization capability of P-DPO still remains robust when no explicit user information is available and only the implicit user model is used.
>
> To summarize, we have conducted ablation studies across multiple dimensions in our experiments: First, we ablate whether to use the user-agnostic loss or not (Figure 5 (b) in Appendix C.3) and found that the user-agnostic loss is critical. Second, we check how our methods perform across different numbers of users (Appendix C.3) and found that our method is effective with varying numbers of users. Third, we ablate on how the number of user clusters (Appendix C.3, Section 5.3) and number of user tokens (Appendix C.3) affect the performance of our proposed algorithm, and found that the performance depends on the number of chosen clusters and it is not necessarily the case that the higher the cluster number the better. Finally, for the PRISM experiments, we ablate on the effectiveness of implicit user model by adding the implicit user model in addition to explicit user information (Section 5.3) and using only the implicit user model (general rebuttal) to verify the necessity of the implicit user model.

---

> ### Author Response · Authors · 2024-11-29
>
> **Weakness 4 & Question 3:** Binary Nature of Preference Modeling and Evaluation Metrics
>
> First, we want to emphasize that **binary preference labels** and **winrate (binary evaluation labels)**, though imperfect, is the **standard data format** and **standard evaluation metric** used in RLHF settings (from the first RLHF paper [7] to its hundreds of follow-up work) and in evaluating language models when one proposes new RLHF algorithms ([3,6,8] and their follow-up work). Though some researchers have been proposing other ways to evaluate LLMs in open-ended generations, none of them is widely adopted. Our main contribution focuses on developing new algorithms in the RLHF settings where one knows additional user information in addition to preference labels instead of proposing new ways of evaluating RLHF algorithms. Thus, in our work, we adopt the standards in the RLHF and language modeling literature. For qualitative (more nuanced) analysis, we have included generations from the personalized LLMs in Appendix D.3 and E.3. There, we look into the actual generations and one can see that the responses generated by the personalized LLMs reflects user preferences.
>
> Beyond this, in our proof-of-concept (Section 5.1), we simulate user preferences solely based on response lengths in order to create a fully controlled setting under which we can derive the expected optimal behavior of personalized LLMs. In this case, when we evaluate the learned personalized LLM, we can go beyond binary evaluation labels (win rate) but actually inspect the model behaviors and check if it can generate the ideal responses.
>
> For our other two experiments—**Instruction Following under Different Preference Profiles** (Section 5.2) and **Personalization on Real-World Preference Dataset with Large User Base** (Section 5.3), the settings are grounded in much more sophisticated and realistic scenarios. Specifically, in Section 5.2, user preferences are based on nuanced attributes such as expertise, informativeness, and style. In Section 5.3, the preferences are based on human users' assessment of the overall quality of responses from various LLMs. These two experiments utilize preference data which captures the diverse and multifaceted nature of user preferences in realistic settings. In addition, as we mentioned, we have looked into the actual generations in Appendix D.3 and E.3 and see that the sampled generations align with user preferences. Therefore, we argue that our experiments have comprehensively validated the capability of our method in modeling sophisticated user preferences, beyond the fully controlled, binary proof-of-concept scenario.
>
> To summarize, in RLHF literature, win-rate is a widely adopted metric to evaluate whether a fine-tuned model better aligns with human preferences compared to another model. Since identifying the ground-truth response is often impractical for open-ended language generation tasks (e.g., the best summarization of a context), win-rates can effectively assess the quality of model generations based on human preference criteria and are frequently used in RLHF studies ([1-5] and their follow-up works). Thus, we used win-rates in our evaluations. However, we agree with the reviewer on the limitations of win-rates. This is precisely why we designed Experiment One with fully simulated preferences, to enable us to rigorously derive the expected optimal behavior of the LLM, and precisely why we looked into the actual generations in Appendix D.3 and E.3. By comparing the learned LLMs with the ideal behavior and looking into the actual generations, we can conduct a more nuanced analysis that goes beyond binary metrics.
>
>
> [1] Ouyang, Long, et al. "Training language models to follow instructions with human feedback." Advances in neural information processing systems 35 (2022): 27730-27744.
>
> [2] Wu, Zeqiu, et al. "Fine-grained human feedback gives better rewards for language model training." Advances in Neural Information Processing Systems 36 (2023): 59008-59033.
>
> [3] Rafailov, Rafael, et al. "Direct preference optimization: Your language model is secretly a reward model." Advances in Neural Information Processing Systems 36 (2024).
>
> [4] Jang, Joel, et al. "Personalized soups: Personalized large language model alignment via post-hoc parameter merging." arXiv preprint arXiv:2310.11564 (2023).
>
> [5] Chakraborty, Souradip, et al. "MaxMin-RLHF: Towards equitable alignment of large language models with diverse human preferences." arXiv preprint arXiv:2402.08925 (2024).
>
> [6] Chiang, Wei-Lin, et al. "Chatbot arena: An open platform for evaluating llms by human preference." arXiv preprint arXiv:2403.04132 (2024).
>
> [7] Stiennon, Nisan, et al. "Learning to summarize with human feedback." Advances in Neural Information Processing Systems 33 (2020): 3008-3021.
>
> [8] Dubois, Yann, et al. "Alpacafarm: A simulation framework for methods that learn from human feedback." Advances in Neural Information Processing Systems 36 (2024).

---

> ### Author Response · Authors · 2024-11-29
>
> **Weakness 5:** Contextual Shifts in User Preferences
>
> When there is a context switch, P-RLHF models can naturally adapt to it through the real-time textual information of that user. When users interact with the language models, their prompts, chat histories (context/topic of the conversation), additional new information about the users can always be added to user textual information in real-time in our method, allowing for generating personalized responses with context switches. In fact, as shown in our Experiment Three (Section 5.3) with the PRISM data, our method works well with changing textual information and change of topics for the users as they interact with the LLMs. If one were to update the implicit user embedding as they collect more preference data, they can retrain the model. Due to the modular nature of our design, instead of needing to fine-tune the LLM, they can just fine-tune the user model. In particular, for cluster-based user models, we can just train the weight vector for users they want to update their embeddings for while holding other parameters frozen. This allows for fast training and update of the model. In summary, when there is a context switch, our method can naturally adapt to it through the textual information part of the user, and if one is willing to update the model, they can do so with a lightweight training run on a small part of the user model. (We added more discussion on the computational cost of the method in the revised manuscript.)

---

### Official Review · Reviewer_oVq3 · 2024-11-03

**Soundness:** 2
**Presentation:** 3
**Contribution:** 2
**Rating:** 3
**Confidence:** 4

**Summary:**

This paper addresses the important challenge of generalizing large language models (LLMs) to capture personalized user preferences. To achieve this, the authors develop several models that incorporate users' implicit preferences, along with a new learning objective aimed at optimizing the reference LLMs for personalization. They compare the proposed method, P-DPO, with the standard DPO in both synthetic and human evaluation settings. In both cases, P-DPO shows advantages. The paper is well-written, with a clearly motivated problem statement. The discussion on the limitations of existing RLHF methods is thorough and insightful.

However, the reviewer has the following concerns about this work.
1. The contribution appears limited. The primary innovation, the new learning objective P-DPO introduced in Section 4.4, facilitates personalization by incorporating user embeddings into the implicit reward model of DPO. However, this approach is not particularly novel, as the integration of user embeddings for personalization is extensively discussed in the recommendation literature [1].
2. The evaluation lacks comprehensiveness. The authors only consider DPO as a baseline and do not include recent work on personalized LLMs, such as [2] and [3]. Additional relevant studies can be found at https://github.com/HqWu-HITCS/Awesome-Personalized-LLM. Furthermore, the experiments overlook the recommendation task, which is a crucial application for personalized LLMs. In the human evaluation, the sample size is limited to at most 20 users and 76 samples, which may lead to biased results and unreliable conclusions.
3. The advantages of integrating both users' explicit and implicit embeddings are not well supported. The authors are suggested to include an ablation study to validate the importance of the learned implicit user embeddings.
4. The motivation for including the user-identifier-agnostic loss term in P-DPO is not clear to the reviewer. The advantage of this design is also not evident.

[1] SSE-PT: Sequential Recommendation Via Personalized Transformer
[2] Factual and Tailored Recommendation Endorsements using Language Models and Reinforcement Learning
[3] Personalized Large Language Models

**Strengths:**

1. The issues related to DPO in personalized modeling are thoroughly discussed.
2. The presentation is easy to follow and well-structured.
3. In both synthetic and human evaluations, P-DPO significantly outperforms DPO.

**Weaknesses:**

For a detailed discussion of the reviewers' concerns, please refer to the summary.

**Questions:**

1. Why include the user-identifier-agnostic loss term in P-DPO?

---

> ### Author Response · Authors · 2024-11-29
>
> We thank the reviewer for their feedback. We appreciate that they found our paper well-written, our motivation clear, and our discussion thorough and insightful. Below we provide responses to the weaknesses and questions:
>
> **Weakness 1:** the contribution appears limited
>
> We emphasize that our work focuses on personalization for **open-ended language generation tasks**, such as summarization, instruction following, and dialog tasks, where comparison-based preferences are an important and commonly collected form of user feedback. Since the golden ground-truth response is challenging or even impossible to obtain for such tasks (e.g. the best summarization of a given context), users’ preferences of one response over the other in pairwise comparisons is a much more practical feedback form which effectively captures users’ preferences. Our P-RLHF framework learns the user embeddings which encode user preferences from such pairwise preference data, using a **personalized DPO learning objective** based on the Bradley-Terry model. To the best of our knowledge, our work is **the first** in the RLHF literature that proposes **new ways to learn user embeddings from preference data** and **integrate such user embeddings for personalized language modeling**, **the first** to provide a scalable way to build such user models, and **the first** to demonstrate the effectiveness of RLHF personalization methods on a large-scale real-world human preference dataset (PRISM, Section 5.3).
>
> In contrast, methods in the recommendation literature [1][2] primarily learn the user embeddings from **user-generated content**, such as user-selected items or user-written reviews, rather than comparison-based preference data. In addition, these methods rely on binary cross entropy loss [1] or collaborative filtering [2] to learn the user embeddings, which are **fundamentally different from our RLHF-based learning objective**. Though both our method and recommendation methods leverage user embeddings, given the fundamental difference in task type, data form, and methodology, we believe that our work is novel and is significantly different from existing personalization methods in the recommendation literature.
>
> Additionally, we highlight the generality of our personalized RLHF framework—the personalized learning objective we propose can be extended to any preference optimization algorithm in RLHF, among which we employed DPO in our experiments due to its wide adoption [4]. Detailed derivations for personalized reward modeling for PPO and personalized IPO are provided in Appendix F.2 and F.3. This demonstrates the **versatility of our personalization techniques across different RLHF methods**.
>
> Given the novelty of learning personalized LLMs from comparison-based preferences and the generality of our framework, we believe our contributions are significant for both RLHF and LLM personalization literature.
>
>
> **Weakness 2:** the evaluation lacks comprehensiveness
>
> As emphasized in our response to Weakness 1, our work focuses on personalization for **open-ended language generation tasks**, where **comparison-based preferences** are a prevalent and practical form of user feedback, and RLHF is widely adopted in such settings to align LLMs with user preferences. In contrast, much of the existing work on personalized LLMs (e.g., [2], [3], and others listed in the Awesome-Personalized-LLM GitHub repository) and methods in the recommendation literature rely on **user-generated content** (e.g. item selections, reviews, demonstrations) to achieve personalization. As a result, these methods are **not applicable** to the comparison-based preference datasets used in our experiments and thus we do not include them as baselines.
>
> For the evaluation in **Personalization on Real-World Preference Dataset with Large User Base** (Section 5.3) using the PRISM dataset, we initially included results from 20 randomly sampled users to manage the computational costs within a limited academic budget. These results demonstrated the effectiveness of our method, achieving win-rates > 50% against strong baselines. In the general rebuttal, we have added results for the evaluation on 70 randomly selected users and 256 conversation turns. The results have shown that on a larger evaluation user base,  our method still outperforms the strong baselines with access to explicit user textual information by achieving above 50% win-rates.
>
> [1] Wu, Liwei, et al. "SSE-PT: Sequential recommendation via personalized transformer." Proceedings of the 14th ACM conference on recommender systems. 2020.
>
> [2] Jeong, Jihwan, et al. "Factual and Tailored Recommendation Endorsements using Language Models and Reinforcement Learning." First Conference on Language Modeling.
>
> [3] Woźniak, Stanisław, et al. "Personalized large language models." arXiv preprint arXiv:2402.09269 (2024).
>
> [4] Dubey, Abhimanyu, et al. "The llama 3 herd of models." arXiv preprint arXiv:2407.21783 (2024).

---

> ### Author Response · Authors · 2024-11-29
>
> **Weakness 3:** the advantages of integrating both users' explicit and implicit embeddings
>
> Our existing experiments have demonstrated **clear advantages** of including both users’ explicit and implicit user embeddings over having just the explicit one: For PRISM experiments (Section 5.3) where we personalize the LLM for dealing with complex user profiles and dialogue topics, we **have compared our P-DPO method with implicit + explicit user embedding against explicit-only** method—vanilla DPO (with explicit user textual information). **L487-489** provides details on the exact setup. As shown in Table 2, our method **wins over** this baseline with just explicit user information.
>
> **Weakness 4 & Question 1:** motivation for including user-agnostic loss
>
> First, the user-agnostic loss trains the generic user embedding $e_0$ so that for new unseen users without user information, our method can generate a generic response that represents the common user preference. Without the user-agnostic loss term, there will be no signal from the learning objective to teach the model to generate good responses based on the generic user embedding for a new unseen user.
>
> Second, empirically, we have conducted an ablation study on removing the user-agnostic loss and found that without it, the model training behaves very unstably for generic user accuracy for reasons as discussed in our first point. As shown in Figure 5 (b) in Appendix C.4, the purple line (with $\alpha=1.0$) is the training trajectory for the accuracy of the generic user without the user-agnostic loss. Thus, **having a user-agnostic loss term is critical**.

---

> ### Author Response · Authors · 2024-12-03
> **Last day for response: Follow up on our rebuttal**
>
> Dear Reviewer oVq3,
>
> As the discussion period is coming to a close, we would greatly appreciate it if you could let us know whether our rebuttal has fully addressed your concerns and questions.
>
> Regarding the four listed concerns and one listed question, we have provided point-by-point responses, including
>
> - We have justified the contribution and novelty of our method, making it clear that **the main focus of our work lies in language model fine-tuning with RLHF** while the cited papers are recommendation or personalization with user-generated content and **not applicable to RLHF**.
> - We have clarified the **fundamental difference** between **learning personalized LLMs under the RLHF setting using pairwise preference data** and personalization in the recommendation setting. Thus, methods in the personalized recommendation literature are not applicable in the RLHF preference data setting, making it impossible to compare our method with them.
> - Our original manuscript contains the experimental details and results which have demonstrated “the advantages of integrating both users' explicit and implicit embeddings.” (L487-489, Table 2)
> - We have explained and verified the necessity and effectiveness of the user-identifier-agnostic loss both through theoretical intuitions and empirical investigations.
>
> Please let us know if there are any additional analyses or results we could provide to further address your concerns. Thank you for your time and consideration.
>
> Best regards, \
> Authors of Submission 12971

---

### Official Review · Reviewer_rjtj · 2024-11-04

**Soundness:** 3
**Presentation:** 3
**Contribution:** 3
**Rating:** 6
**Confidence:** 3

**Summary:**

The paper introduces Personalized-RLHF (P-RLHF), an innovative framework designed to enhance personalized content generation in large language models (LLMs) by leveraging a lightweight user model. Its key contributions include:
Scalability: P-RLHF allows LLMs to efficiently generate personalized content while accommodating a growing user base.
Preference Handling: The framework effectively processes both explicit user preferences, provided as textual input, and implicit preferences derived from feedback data.
Reduced User Burden: It alleviates the requirement for users to fully articulate their preferences, addressing the challenges of obtaining detailed user input in practical applications.

Overall, this paper is easy to follow, but there are issues such as a lack of innovation, no released code, and an excess of symbols. However, overall, this is a ready piece of work.

**Strengths:**

* The structure of this article is well-organized and clearly articulated, making it easy for readers to follow the flow of ideas and concepts presented throughout the text. Each section is logically arranged, allowing for a seamless understanding of the material.

* The section on RLHF is detailed, with a clear progression from general RLHF to the specifically designed personalized DPO. This work considers different user preferences from multiple angles and granularities. It clearly explains the differences between P-RLHF and Vanilla DPO to clarify the motivation.

* The work uses the TL;DR dataset, which has Reddit posts, summaries, and worker IDs. It conducts an experiment where most workers (70%) prefer longer responses and a smaller group (30%) prefers shorter ones. The top 10 workers with the most annotations were included for training. After the experiment, some workers liked shorter responses while others liked longer ones. Overall, this shows a strong effort and good understanding of data analysis, which is impressive.

* There is a solid exploration of how personalized LLMs can adapt to different user preferences through P-DPO fine-tuning. The study found that the personalized model can generate longer or shorter responses based on users' implicit preferences, demonstrating its flexibility and adaptability. The experiment also compared the performance of P-DPO using generic user embeddings with traditional vanilla DPO, showing that the personalized model excels in understanding and responding to user needs, validating its effectiveness in practical applications.

**Weaknesses:**

* It seems that this article does not have released code. I'm not sure if I just couldn't find it, but without released code, reproducibility cannot be guaranteed.

* Like other RL works, the notation in this paper is too numerous and complex. Although the structure and presentation of the article are good, it somewhat hinders the readers' understanding. Perhaps a table to organize the notations could be helpful.

* There are concerns regarding the effectiveness of not training an additional reward model and relying solely on the user model.

**Questions:**

* Does this work have released code and dataset?
* There are works have been done on personalized user preferences using RLHF. How does this one differ from them, and is there a comparison used as a baseline?

---

> ### Author Response · Authors · 2024-11-29
>
> We thank the reviewer for their feedback. We appreciate that they found our presentation clear, our data analysis impressive, and our method effective. Below we provide responses to the weaknesses and questions:
>
> **Weakness 1 & Question 1:** does this work have released code and dataset?
>
> The datasets used in our study—TL;DR, Personalized Soups, and PRISM—are all publicly available. We provide the link to them as below:
> - TL;DR: https://huggingface.co/datasets/openai/summarize_from_feedback
> - Personalized Soups: https://github.com/joeljang/RLPHF
> - PRISM: https://huggingface.co/datasets/HannahRoseKirk/prism-alignment
>
> We will add the link to our code in the final de-anonymized submission. The anonymized code can be found in this repository: https://anonymous.4open.science/r/Personalized_RLHF-DD4C/
>
>
> **Weakness 2:** the notation is numerous and complex
>
> Thank you for your feedback and suggestions on improving the readability of our paper. We tried to streamline the notations where possible to improve clarity, and if space permits, we will include a table summarizing all the notations in the camera-ready version.
>
>
> **Weakness 3:** concerns regarding not training an additional reward model
>
> We used DPO as the RLHF method to train the personalized LLMs. In the DPO formulation, the personalized reward $r(x, y, u^t, u^p) = \beta \log \frac{\pi_\text{P}(y | x, u^t, u^p)}{\pi^\text{SFT}(y |x)}$ is implicitly defined in the DPO learning objective, where $x, y, u^t, u^p$ stand for the prompt, the response, the user textual information, and the unique user identifier. By leveraging the optimal policy solution and comparison-based preference data [1], DPO eliminates the need for training a separate reward model and the two-stage RL training procedure typical of methods like PPO, achieving the same policy optimization goal while significantly reducing computational cost. Hence, the reward is optimized during DPO training, albeit not as a separate component.
>
> In **Generation with Conflicting Preferences** experiment (Section 5.1), we analyzed the DPO reward accuracy to verify whether our method is able to correctly assign higher rewards to user-preferred responses. Our results show that P-DPO significantly outperforms the non-personalized, vanilla DPO baseline in the reward space, achieving over 90% accuracy on seen users compared to the baseline’s 59% accuracy. Further details can be found in Appendix C.3.
>
>
> **Question 2:** difference from existing personalized user preferences using RLHF
>
> Existing studies on personalizing LLM outputs with RLHF often rely on predefined preference dimensions, learning separate reward models or LLM policies for these dimensions, and then personalizing responses by adjusting reward weights [2] or merging LLMs based on specific preference choices [3]. Our work differs in two key ways: (1) we directly learn personalized LLMs from user information and personalized feedback data, without relying on predefined preference dimensions; and (2) our method does not require training multiple LLMs or reward models. Instead, we use a lightweight user model to augment a base LLM, significantly improving the scalability of the personalization method.
>
> For **Generation with Conflicting Preferences** (Section 5.1) and **Instruction Following under Different Preference Profiles** (Section 5.2) experiments, we directly compare our learned personalized LLMs against ground-truth preferences since they are known. In Section 5.1, we verify the learned personalized LLM against the expected optimal behavior derived mathematically from ground-truth preferences, while in Section 5.2, we compare against the responses generated from SFT prompted with ground-truth preference profiles.
>
> For the PRISM dataset used in **Personalization on Real-World Preference Dataset with Large User Base** (Section 5.3) experiment, human users provided preferences based on the overall quality of LLM responses, without relying on predefined preference dimensions. As a result, existing methods that depend on predefined dimensions could not serve as baselines. Instead, we compared our method against DPO with access to user and preference information, which we believe is a strong and relevant baseline.
>
>
> [1] Rafailov, Rafael, et al. "Direct preference optimization: Your language model is secretly a reward model." Advances in Neural Information Processing Systems 36 (2024).
>
> [2] Wu, Zeqiu, et al. "Fine-grained human feedback gives better rewards for language model training." Advances in Neural Information Processing Systems 36 (2023): 59008-59033.
>
> [3] Jang, Joel, et al. "Personalized soups: Personalized large language model alignment via post-hoc parameter merging." arXiv preprint arXiv:2310.11564 (2023).

---

### Author Response · Authors · 2024-11-29

We thank all the reviewers for their thorough reviews and constructive feedback. We are encouraged to see that the reviewers recognize the importance of our topic (Reviewers oVq3, hQjp), find our method well-motivated (Reviewer hQjp), effective (Reviewers rjtj, oVq3, uiaG), and generally applicable (Reviewer hQjp), recognize the clarity and structure of our presentation (Reviewers rjtj, oVq3, uiaG), and acknowledge the significance of our contribution (Reviewer uiaG). Below, we first present results from an additional set of experiments on the PRISM dataset (Section 5.3) to further demonstrate the effectiveness and necessity of the implicit user model in personalized LLMs. We then address the specific questions from each reviewer individually.

For the **Personalization on Real-World Preference Dataset with Large User Base** experiment (Section 5.3), we add an additional variant of our P-DPO method—**P-DPO with only the implicit user model**, for cluster-based preference with number of clusters K=10. We compare the newly trained P-DPO model to the same two baselines—vanilla DPO with explicit user information and the original chosen responses in the dataset, and evaluate on both the original evaluation set of 20 users (76 conversation turns) reported in Section 5.3 and a larger evaluation set of 70 users (256 conversation turns). The results are shown in the table below:

|  |  |  num_users = 20 | num_users = 70 |
|----------|----------|----------|----------|
| per-sample win rate | vs. vanilla DPO         |  51.31  |  65.38 |
| per-sample win-rate | vs. chosen response |  75.00 |  76.95  |
| per-user win-rate | vs. vanilla DPO              |  50.00 |  65.56  |
| per-user win-rate | vs. chosen response      |  77.75  |  78.52  |

The results demonstrate that the personalization capability of P-DPO remains effective and robust without access to explicit user information, with the implicit user model successfully capturing user preferences directly from the personalized feedback in the PRISM dataset. Furthermore, when tested on a larger user base (70 users), P-DPO continues to outperform the two strong baselines, achieving win rates > 50% as in the 20-user evaluation.

---

### Meta-Review · Area_Chair_JeUW · 2024-12-22

**Metareview:**

The paper introduces Personalized-RLHF (P-RLHF), an new framework designed to enhance personalized content generation in large language models (LLMs) by leveraging a lightweight user model. The proposed method allows the LLMs to efficiently generate personalized content, and processes both the explicit and implicit user preferences. The authors have performed experiments on real dataset to demonstrate the effectiveness of the proposed method.

Overall, this paper is clearly written, and the authors have introduced some new ideas. However, for this work, the experimental analysis is not sufficient. Firstly, the authors do not compare the proposed method with existing personalized LLM methods. Secondly, how the proposed model can handle the both the user implicit and explicit preference is not well studied. How explicit preference and implicit preference will impact the performance of the proposed model should also be well studied. Besides showing some metric numbers, the authors are also recommended to present some case studies to demonstrate the effectiveness of the proposed method.

**Additional Comments On Reviewer Discussion:**

In the rebuttal, the authors have addressed some concerns of the reviewers, e.g., providing the link to the source codes, explaining the notations in the papers, discussing the differences between this paper and existing personalized LLMs. Moreover, the authors also provide additional ablation studies about the robustness of the proposed model, when no explicit user information is available.

---

### Decision · Program_Chairs · 2025-01-22

Reject